# Noisy Pairwise-Comparison Random Search for Smooth Nonconvex Optimization

**Taha El Bakkali El Kadi** [1]  **Rayane Bouftini** [1]  **Qiuyi Zhang** [2]
**Omar Saadi** [1]

## Abstract

We study smooth nonconvex optimization using only noisy pairwise comparisons, without access to gradients or function values. We propose Noisy-Comparison Random Search (NCRS), a simple direct-search method that samples random directions and performs accept/reject updates from comparison feedback. Under a low-dimensional active-subspace structure, NCRS adapts to the intrinsic dimension $k \leq d$ rather than the ambient dimension $d$. For a uniform-margin comparison oracle with advantage $p$, NCRS achieves $\epsilon$-first-order stationarity with comparison complexity $\mathcal{O}(k/(p^2\epsilon^2))$. We also introduce a gap-dependent confidence model, where comparison reliability decreases as the objective-value gap between the two candidates becomes small, and analyze a confidence-weighted voting variant of NCRS. For this oracle, the method achieves $\epsilon$-first-order stationarity with total comparison complexity $\mathcal{O}(k^2/\epsilon^4)$. These results provide intrinsic-dimension convergence guarantees for noisy comparison-based random search in smooth nonconvex optimization.

## 1. Introduction

We study smooth nonconvex optimization of an objective $f : \mathbb{R}^d \to \mathbb{R}$ under a feedback model that is increasingly common in modern machine learning: rather than observing gradients or function values, the optimizer only receives noisy comparisons between candidate solutions. Given two points $x, y \in \mathbb{R}^d$, the oracle returns a possibly noisy ordinal signal indicating which point has the smaller value of $f$. The optimizer observes neither $f(x)$, $f(y)$, nor gradient information (Zhang & Ying, 2025; Saha et al., 2025; Tang

et al., 2024). The goal is to find an $\varepsilon$-first-order stationary point, namely a point $x$ satisfying $\|\nabla f(x)\|_2 \leq \varepsilon$, using as few comparison queries as possible.

This setting arises naturally in preference-driven applications. In reinforcement learning from human feedback, model outputs are compared according to human preferences; in recommender systems and search-result ranking, user interactions reveal relative preferences among items; and in human-in-the-loop generation, candidate outputs such as images or responses are often evaluated through pairwise judgments.

In the context of language-model alignment, ComPO (Chen et al., 2025) recently proposed a zeroth-order preference-alignment method based on comparison oracles, further illustrating the relevance of comparison feedback in modern machine-learning applications.

Prior work has shown that ordinal feedback can be surprisingly informative when the comparison oracle is exact (Golovin et al., 2020; Tang et al., 2024). In this deterministic setting, ranking-based zeroth-order methods can extract optimization-relevant information from orderings alone and obtain convergence guarantees for smooth nonconvex objectives. In particular, Tang et al. (2024) achieve a comparison-query complexity of order $O(d/\varepsilon^2)$ for finding an $\varepsilon$-first-order stationary point. This is remarkable because the same $O(d/\varepsilon^2)$ scaling is also achieved by classical smooth nonconvex zeroth-order methods under the stronger value-oracle model, where exact function values are observed.

However, exact comparisons are often unrealistic in preference-driven applications. When two candidates have similar objective values, their ordering may be ambiguous and the observed preference may be unreliable. Thus, the optimizer must extract descent information from comparisons that are not only ordinal, but also noisy and potentially weak when the objective gap is small. This motivates the study of noisy comparison oracles for smooth nonconvex optimization.

A central difficulty in comparison-based optimization is its severe information bottleneck relative to first-order meth-

---

[1]College of Computing, UM6P [2]Google DeepMind. Correspondence to: Taha El Bakkali El Kadi <taha.elbakkali@um6p.ma>.

*Proceedings of the 43$^{rd}$ International Conference on Machine Learning*, Seoul, South Korea. PMLR 306, 2026. Copyright 2026 by the author(s).

ods. A gradient query directly reveals a full vector in $\mathbb{R}^d$, providing an explicit local descent direction. By contrast, a comparison query provides only relative information about two queried points. In the simplest deterministic case, this information reduces to a single bit indicating whether $f(y) < f(x)$. Thus, even in the absence of noise, the optimizer must infer descent information from highly compressed ordinal observations rather than observing a descent direction directly. This limitation becomes especially pronounced in high dimension: random search directions probe the gradient only through one-dimensional projections, leading to an unavoidable dependence on the ambient dimension $d$ (Jamieson et al., 2012).

Recent work has begun to address more realistic noisy preference models. In the smooth convex and strongly convex settings, Saha et al. (2025) study a general dueling framework in which the expected comparison outcome is governed by a transfer function $\rho$ of the objective gap. In this model, the preference signal can become weaker as the two queried candidates become closer in objective value, and the resulting query complexity depends polynomially on the local flatness degree of the transfer function. This highlights a key difficulty of noisy comparison feedback: when the objective gap between two candidates is small, the observed preference may carry only weak directional information. While this phenomenon already arises near optimality in convex optimization, it is particularly important in smooth nonconvex optimization, where convergence is certified by a small gradient norm. As $\|\nabla f(x)\|_2$ decreases, nearby perturbations tend to have similar objective values, making noisy comparisons weakly informative precisely when stationarity must be established.

Beyond noise, high dimensionality raises an additional challenge. Modern machine learning models often have enormous ambient dimension, while the objective may vary meaningfully only along a much lower-dimensional structure. This motivates the distinction between the ambient dimension $d$ and an intrinsic dimension $k$, corresponding to the dimension of an active subspace along which the objective exhibits most of its variation. Classical zeroth-order and comparison-based methods typically incur a cost depending on $d$, which can be prohibitive when $d$ is large. A natural question is therefore whether comparison-based random search can exploit such intrinsic structure while remaining robust to noisy ordinal feedback.

### 1.1. Intrinsic Dimension via a Ridge Model

A common way to model intrinsic dimension is to assume that, although the ambient dimension is large, the objective varies only through a few effective directions. This viewpoint is classical in active-subspace methods, which are motivated by the observation that many multivariate

engineering functions depend primarily on a small number of important directions in the input space (Constantine et al., 2014). Related evidence appears in language-model fine-tuning, where low-dimensional reparameterizations can often achieve performance comparable to full-space optimization (Aghajanyan et al., 2021).

Motivated by these observations, we adopt the ridge model

$$f(x) = g(Ax),$$

where $A \in \mathbb{R}^{k \times d}$ has full row rank $k \leq d$, and $g : \mathbb{R}^k \to \mathbb{R}$ is a low-dimensional function. In this model, $f$ depends on $x$ only through the $k$-dimensional representation $Ax$, and is invariant along directions in $\ker(A)$. Thus, $k$ captures the intrinsic dimension of the objective, while $d$ is the ambient dimension.

This ridge assumption is not meant to claim that practical objectives exactly satisfy a fixed low-dimensional representation. Rather, it provides a tractable analytical abstraction of intrinsic dimension.

### 1.2. Stochastic Sign-Comparison Oracle

We assume access to a stochastic pairwise-comparison oracle $R : \mathbb{R}^d \times \mathbb{R}^d \to \{-1, +1\}$, intended to return the sign of the true difference $f(x) - f(y)$ but may be noisy. We measure its reliability through:

$$\Pr\big(R(x,y) = \operatorname{sign}(f(x) - f(y))\big).$$

**Uniform-margin comparisons.** A classical model is the *uniform-margin* regime, which assumes that the oracle remains uniformly better than random guessing, independently of the queried pair. While convenient for analysis, this assumption can be optimistic in applications, since comparisons are typically most ambiguous when $|f(x) - f(y)|$ is small near ties.

**Assumption 1.1.** There exists $p \in (0, \frac{1}{2}]$ such that for all $x, y \in \mathbb{R}^d$ with $f(x) \neq f(y)$,

$$\Pr\big(R(x,y) = \operatorname{sign}(f(x) - f(y))\big) \ \geq \ \tfrac{1}{2} + p,$$

where the oracle output is $R(x,y) \in \{-1, +1\}$. When $f(x) = f(y)$, the oracle output may be arbitrary.

A convenient scalar summary of the comparison quality is the signed product $R(x,y) \operatorname{sign}(f(x) - f(y))$. In the noiseless case this equals 1 for every queried pair, since $R(x,y)$ always matches the true ordering. If we suppose that the oracle is non-informative and, given $(x,y)$, returns a fair random sign $R(x,y) \in \{-1, +1\}$ independent of the true ordering, then:

$$\mathbb{E}[R(x,y) \operatorname{sign}(f(x) - f(y)) \mid x, y] = 0.$$

*Table 1.* Comparison of comparison-based optimization algorithms.

| Algorithm | Objective / oracle model | Complexity / guarantee | Metric |
|---|---|---|---|
| **Jamieson et al.** (Jamieson et al., 2012) | Smooth Convex, noiseless comparisons | $\Omega(d\log(1/\epsilon))$ | $\mathbb{E}[f(x)] - f^\star$ |
| **GLD** (Golovin et al., 2020) | Monotone transform of smooth + strongly convex; noiseless comparison, intrinsic dim $k$ | $\mathcal{O}(k\log(1/\epsilon))$ | $\mathbb{E}[f(x)] - f^\star$ |
| **Saha et al.** (Saha et al., 2021) | Smooth convex; uniform-margin noise, advantage $p$ | $\tilde{\mathcal{O}}(d/(p^2\epsilon))$ | $\mathbb{E}[f(x)] - f^\star$ |
| **Saha et al.** (Saha et al., 2021) | Strongly convex; uniform-margin noise, advantage $p$ | $\tilde{\mathcal{O}}((d/p^2)\log(1/\epsilon))$ | $\mathbb{E}[f(x)] - f^\star$ |
| **Saha et al.** (Saha et al., 2025) | Smooth Convex; margin-dep noise model, noise power $q$ | $\tilde{\mathcal{O}}(d^{2q+1}/\epsilon^{4q})$ | $\mathbb{E}[f(x)] - f^\star$ |
| **ZO-RankSGD** (Tang et al., 2024) | Smooth nonconvex; noiseless ranking | $\mathcal{O}(d/\epsilon^2)$ | $\mathbb{E}\|\nabla f(x)\|_2$ |
| **Wang et al.** (Wang et al., 2025) | Smooth nonconvex; sign of two noisy evaluations, variances $\sigma, \sigma_f$ | $\tilde{\mathcal{O}}(d/\epsilon^2)$ for $\sigma + \sigma_f = \tilde{\mathcal{O}}(\epsilon)$ | $\mathbb{E}\|\nabla f(x)\|_2$ |
| **NCRS (Our Work)** | Smooth nonconvex, intrinsic dim $k$; uniform-margin noise, advantage $p$ | $\mathcal{O}(k/(p^2\epsilon^2))$ | $\mathbb{E}\|\nabla f(x)\|_2$ |
| **NCRS+Vote (Our Work)** | Smooth nonconvex, intrinsic dim $k$; confidence model | $\mathcal{O}(k^2/\epsilon^4)$ | $\mathbb{E}\|\nabla f(x)\|_2$ |

By denoting $Q_{x,y} := \mathbb{E}[R(x,y)\operatorname{sign}(f(x) - f(y)) \mid x,y]$, we remark that:

$$Q_{x,y} = 2\Pr(R(x,y) = \operatorname{sign}(f(x) - f(y)) \mid x,y) - 1.$$

Assumption 1.1 enforces a uniform positive bias toward the correct ordering,

$$\mathbb{E}[R(x,y)\operatorname{sign}(f(x) - f(y)) \mid x,y] \geq 2p,$$

even when $|f(x) - f(y)|$ is arbitrarily small.

**Gap-dependent confidence comparisons (Our model).**
To reflect the fact that comparisons are hardest near ties, we consider an enriched oracle that outputs a signed confidence score:
$$\tilde{R} : \mathbb{R}^d \times \mathbb{R}^d \to [-1, 1].$$

We interpret $\tilde{R}(x,y) > 0$ as "$y$ is preferred to $x$", $\tilde{R}(x,y) < 0$ as "$x$ is preferred to $y$", and $|\tilde{R}(x,y)|$ as a confidence level. Define $\Delta(x,y) := f(x) - f(y)$.

**Assumption 1.2.** For any queried pair $x, y \in \mathbb{R}^d$ with $\Delta(x,y) \neq 0$, the oracle returns $\tilde{R}(x,y) \in [-1,1]$ such that

$$\begin{cases} \mathbb{E}\big[\operatorname{sign}(\Delta(x,y))\tilde{R}(x,y) \mid x,y\big] \geq \rho(|\Delta(x,y)|), \\ \mathbb{E}\big[\tilde{R}(x,y)^2 \mid x,y\big] \leq C\rho(|\Delta(x,y)|), \end{cases}$$

for some constant $C \in [1,\infty)$, where $\rho : \mathbb{R}_+ \to [0,1]$ is nondecreasing with $\rho(0) = 0$. Moreover, there exist constants $c > 0$ and $r > 0$ such that for all $t \in [0, r]$, we have $\rho(t) \geq ct$.

*Remark* 1.3 (Why Assumption 1.2 is more realistic than uniform margin). Assumption 1.1 is gap-independent: it enforces a uniform positive correlation with the true ordering even when $|\Delta(x,y)|$ is arbitrarily small. In contrast, Assumption 1.2 makes the guaranteed alignment scale with

the gap through $\rho(|\Delta(x,y)|)$, allowing the signal to vanish near ties and to increase as one point becomes clearly better.

The second-moment bound encodes soft abstention. Since $|\tilde{R}(x,y)| \leq 1$, small $\rho(|\Delta|)$ forces the score to be typically close to 0. For any $\eta \in (0,1]$, Markov's inequality yields:

$$\Pr\big(|\tilde{R}(x,y)| \geq \eta \mid x,y\big) \leq \frac{C}{\eta^2}\rho(|\Delta(x,y)|).$$

Thus, when $|\Delta(x,y)|$ is small, the oracle rarely outputs a large-magnitude high-confidence preference.

**Connection to classical link functions.** A standard model for noisy comparisons assumes a probabilistic link $\sigma : \mathbb{R} \to (0,1)$ such that $\Pr\big(B(x,y) = +1 \mid x,y\big) = \sigma\big(\Delta(x,y)\big)$, where $B(x,y) \in \{\pm 1\}$ and $\Delta(x,y)$ is the signed gap. This induces the signed score $\rho(t) := \mathbb{E}[B(x,y) \mid \Delta(x,y) = t] = 2\sigma(t) - 1 \in [-1,1]$. We work with a score $\tilde{R}(x,y) \in [-1,1]$ satisfying the mean-consistency condition $\mathbb{E}[\tilde{R}(x,y) \mid x,y] = \rho(\Delta(x,y))$. If $\sigma$ is nondecreasing and satisfies the symmetry $\sigma(-t) = 1 - \sigma(t)$, then $\rho$ is odd and nondecreasing. Moreover, if $\sigma$ is differentiable at 0 with $\sigma'(0) > 0$, then $\rho$ is locally linear near the origin, which implies Assumption 1.2 with constants controlled by $\sigma'(0)$. For example, the Logistic/Bradley–Terry link $\sigma(t) = \big(1 + \exp(-t/\beta)\big)^{-1}$ satisfies $\rho(t) = 2\sigma(t) - 1 = \tanh\big(t/(2\beta)\big)$ and $\rho'(0) = 1/(2\beta)$. The Probit/Thurstone link $\sigma(t) = \Phi(t/\beta)$ yields $\rho(t) = 2\Phi(t/\beta) - 1 = \operatorname{erf}\big(t/(\beta\sqrt{2})\big)$ and $\rho'(0) = \sqrt{2/\pi}\,1/\beta$. Appendix F provides the full derivations and further link functions.

### 1.3. Our Contributions

We study smooth nonconvex optimization when feedback is restricted to noisy pairwise comparisons and the objective

admits a low-dimensional active-subspace structure. We introduce a simple comparison-based random search method and analyze it under two oracle models of increasing realism. At a high level, our contributions are:

**(1) Intrinsic-dimension stationarity under noisy comparisons.** We propose *Noisy-Comparison Random Search* (NCRS; Algorithm 1), a direct-search method that samples a random direction $s_t$ and performs an improve-or-stay update from a single noisy comparison $R(\theta^t, \theta^t + \alpha_t s_t)$. Under a uniform-margin ranking oracle (Assumption 1.1), we show that for smooth nonconvex objectives with a $k$-dimensional active-subspace structure, $\frac{1}{T}\sum_{t=1}^{T}\mathbb{E}\|\nabla f(\theta^t)\|_2 \le \varepsilon$ with $T = \mathcal{O}(k/(p^2\varepsilon^2))$. The key observation is that for ridge-type objectives $f(x) = g(Ax)$, if $P := A^\top(AA^\top)^{-1}A$ denotes the orthogonal projector onto the $k$-dimensional active subspace, then NCRS' accept/reject decision depends only on the projected direction $Ps_t$, so the ambient dimension $d$ is replaced by the intrinsic dimension $k$.

**(2) Gap-dependent confidence comparisons.** Uniform-margin models enforce a gap-independent advantage, which can be overly optimistic near ties. We introduce a gap-dependent *confidence* oracle (Assumption 1.2) that returns a signed score $\tilde{R}(x, y) \in [-1, 1]$ whose alignment and variance are controlled by a function $\rho(|f(x) - f(y)|)$ of the function gap. We analyze a confidence-weighted vote variant of NCRS (Algorithm 2) and show that in the same smooth nonconvex active-subspace setting it achieves $\frac{1}{T}\sum_{t=1}^{T}\mathbb{E}\|\nabla f(\theta^t)\|_2 \le \varepsilon$ with total comparison complexity $NT = \mathcal{O}(k^2/\varepsilon^4)$. The additional $k/\varepsilon^2$ factor captures the near-stationary regime where comparisons become less informative as $|f(x) - f(y)|$ shrinks.

**(3) Empirical validation on intrinsic-dimension and preference-based tasks.** We empirically validate NCRS on masked language model fine-tuning and preference-based RL benchmarks, probing intrinsic-dimension effects and comparing against standard zeroth-order and policy-optimization baselines.

## 2. Uniform-Margin Ranking Oracle: NCRS Analysis

### 2.1. Convergence Analysis of NCRS for Ridge Objectives $f(x) = g(Ax)$

**Low-dimensional structure with monotone observation.** We consider objectives that depend on the decision variable $x \in \mathbb{R}^d$ only through a $k$-dimensional linear embedding, with $k \le d$. A convenient way to formalize this is via a composite representation $f(x) = g(Ax)$, where $A \in \mathbb{R}^{k \times d}$ has rank $k \le d$ and $g : \mathbb{R}^k \to \mathbb{R}$ is a low-dimensional function. We further assume that $f$ is $L_f$-smooth.

**Stochastic ranking oracle.** We assume access to a noisy pairwise-comparison oracle for $f$ satisfying the uniform-margin condition of Assumption 1.1.

**Noisy-Comparison Random Search.** We introduce *Noisy-Comparison Random Search* (NCRS), a simple comparison-based direct-search method that uses one oracle query per iteration. At iterate $\theta^t$, NCRS samples a Gaussian direction $s_t \sim \mathcal{N}(0, I_d)$ and forms the candidate point $\theta^t + \alpha_t s_t$. It then queries the ranking oracle and applies an *improve-or-stay* rule: the candidate is accepted if it is ranked no worse than the current iterate; otherwise the iterate is left unchanged.

---

**Algorithm 1** Noisy-Comparison Random Search (NCRS)

---

1: **Input:** $\theta^1 \in \mathbb{R}^d$, step sizes $(\alpha_t)_{t \ge 1}$
2: **for** $t = 1, 2, \dots$ **do**
3:      Sample $s_t \sim \mathcal{N}(0, I_d)$
4:      Query $R(\theta^t, \theta^t + \alpha_t s_t) \in \{+1, -1\}$
5:      **if** $R(\theta^t, \theta^t + \alpha_t s_t) = +1$ **then**
6:          $\theta^{t+1} = \theta^t + \alpha_t s_t$
7:      **else**
8:          $\theta^{t+1} = \theta^t$
9:      **end if**
10: **end for**

---

**NCRS automatically adapts to the intrinsic subspace.** In the ridge model $f(x) = g(Ax)$, the objective is invariant to motions in $\ker(A)$: $f(x + v) = f(x)$ for all $v \in \ker(A)$. Thus, any component of a perturbation that lies in $\ker(A)$ is *invisible* to the objective and cannot affect the accept/reject decision of NCRS.

Formally, since $\ker(A) = \mathrm{range}(A^\top)^\perp$, we have the orthogonal decomposition $\mathbb{R}^d = \mathrm{range}(A^\top) \oplus \ker(A)$. For any $x \in \mathbb{R}^d$, let $p = Px$ be the orthogonal projection of $x$ onto $\mathrm{range}(A^\top)$. Since $p \in \mathrm{range}(A^\top)$ and $x - p \in \ker(A)$, it holds that:

$$\begin{cases} \exists u \in \mathbb{R}^k, \ Px = A^\top u, \\ Ax = APx = AA^\top u. \end{cases}$$

Since we assume $\mathrm{rank}(A) = k$, the matrix $AA^\top \in \mathbb{R}^{k \times k}$ is invertible, and hence $Px = A^\top(AA^\top)^{-1}Ax$ and $P := A^\top(AA^\top)^{-1}A$ is the orthogonal projector onto $\mathrm{range}(A^\top)$. For any $x \in \mathbb{R}^d$, $\alpha > 0$, and direction $s \in \mathbb{R}^d$,

$$f(x + \alpha s) = g(Ax + \alpha APs) = f(x + \alpha Ps).$$

In particular, the true comparison between $x$ and $x + \alpha s$ depends only on $Ps$:

$$\mathrm{sign}\big(f(x + \alpha s) - f(x)\big) = \mathrm{sign}\big(f(x + \alpha Ps) - f(x)\big).$$

Therefore, although NCRS samples $s_t \sim \mathcal{N}(0, I_d)$ in $\mathbb{R}^d$, its oracle query and accept/reject decision depend only on

the projected direction $Ps_t \in \text{range}(A^\top)$. Equivalently, NCRS behaves as a random-direction comparison search on the k-dimensional active subspace $\text{range}(A^\top)$: components in $\ker(A)$ are invisible to the objective and play no role in the update. This is exactly what yields the $k$-dependence in our convergence bounds, despite NCRS never observing $A$ or the intrinsic dimension $k$.

The next lemma establishes a one-step expected descent inequality for NCRS in the ridge model, with dependence on the intrinsic dimension $k$.

---

**Lemma 2.1.** *Assume that $f(x) = g(Ax)$ for some matrix $A \in \mathbb{R}^{k \times d}$ with full row rank $\text{rank}(A) = k \leq d$ and some function $g : \mathbb{R}^k \to \mathbb{R}$, and that $f$ is $L_f$-smooth. Let $(\theta^t)$ be the iterates produced by Algorithm 1, and suppose the ranking oracle satisfies Assumption 1.1 with parameter $p$. Then for all $t \geq 1$,*

$$p\alpha_t \sqrt{\frac{2}{\pi}} \mathbb{E} \|\nabla f(\theta^t)\|_2 \leq \mathbb{E}[f(\theta^t) - f(\theta^{t+1})] + \frac{L_f k \alpha_t^2}{2}.$$

---

For fixed $T \geq 1$, by averaging the inequality of Lemma 2.1 over the iterations 1 to $T$, while using constant step-size $\frac{\alpha_0}{\sqrt{kT}}$ with $\alpha_0 > 0$, we obtain the following result.

---

**Theorem 2.2.** *Assume that $f(x) = g(Ax)$ for some matrix $A \in \mathbb{R}^{k \times d}$ with full row rank $\text{rank}(A) = k \leq d$ and some function $g : \mathbb{R}^k \to \mathbb{R}$, and that $f$ is $L_f$-smooth and bounded below, i.e., $f^\star := \inf_{x \in \mathbb{R}^d} f(x) > -\infty$. Let $(\theta^t)$ be generated by Algorithm 1, and suppose the ranking oracle satisfies Assumption 1.1 with parameter $p \in (0, \frac{1}{2}]$. Define $\Delta f := f(\theta^1) - f^\star$ and let $T \geq 1$. If NCRS is run with constant stepsizes $\alpha_t = \frac{\alpha_0}{\sqrt{kT}}$ for some $\alpha_0 > 0$, then we have:*

$$\frac{1}{T} \sum_{t=1}^{T} \mathbb{E}[\|\nabla f(\theta^t)\|_2] \leq \frac{1}{p} \sqrt{\frac{\pi}{2}} \left( \frac{\Delta f}{\alpha_0} + \frac{L_f \alpha_0}{2} \right) \sqrt{\frac{k}{T}}.$$

---

*Remark* 2.3. Theorem 2.2 implies that:

$$\frac{1}{T} \sum_{t=1}^{T} \mathbb{E}[\|\nabla f(\theta^t)\|_2] = \mathcal{O}\left( \frac{1}{p} \sqrt{\frac{k}{T}} \right).$$

Thus, by choosing $T = \mathcal{O}\left( \frac{k}{p^2 \varepsilon^2} \right)$, we can ensure that $\frac{1}{T} \sum_{t=1}^{T} \mathbb{E}[\|\nabla f(\theta^t)\|_2] \leq \varepsilon$. In particular, in the deterministic setting $p = \frac{1}{2}$ this recovers an $\mathcal{O}(k/\varepsilon^2)$ evaluation complexity. This bound improves over the classical smooth nonconvex zeroth-order rate $\mathcal{O}(d/\varepsilon^2)$ by replacing the ambient dimension $d$ with the intrinsic dimension $k$.

## 3. Gap-Dependent Confidence Oracle: NCRS Analysis with Confidence-Weighted Vote

Throughout this section we assume an exact ridge structure: there exist $k \leq d$, a full row-rank matrix $A \in \mathbb{R}^{k \times d}$ with $\text{rank}(A) = k$, and a function $g : \mathbb{R}^k \to \mathbb{R}$ such that $f(x) = g(Ax)$ for all $x \in \mathbb{R}^d$. We further assume that $f$ is $L_f$-smooth on $\mathbb{R}^d$ and bounded below by $f^\star$. We now state the *gap-dependent confidence* comparison model.

**Gap-dependent confidence oracle.** For any pair $(\theta^1, \theta^2) \in \mathbb{R}^d \times \mathbb{R}^d$, the oracle returns a signed confidence score $\tilde{R}(\theta^1, \theta^2) \in [-1, 1]$ interpreted as:

$$\begin{cases} |\tilde{R}(\theta^1, \theta^2)| & \text{is the confidence level of the answer,} \\ \tilde{R}(\theta^1, \theta^2) > 0 & \text{means the oracle prefers } \theta^2 \text{ over } \theta^1, \\ \tilde{R}(\theta^1, \theta^2) < 0 & \text{means the oracle prefers } \theta^1 \text{ over } \theta^2. \end{cases}$$

The oracle randomness is understood to be conditional on the queried pair $(\theta^1, \theta^2)$, and it satisfies Assumption 1.2.

**Algorithm.** At iteration $t$, sample $s_t \sim \mathcal{N}(0, I_d)$, form the candidate $\theta^t + \alpha_t s_t$, collect $N$ i.i.d. oracle outcomes on the pair, aggregate by the sign of their sum. This is a confidence-weighted vote. Accept if the aggregate prefers the candidate.

---

**Algorithm 2** NCRS with confidence-weighted vote

1: **Input:** initial point $\theta^1 \in \mathbb{R}^d$, stepsizes $(\alpha_t)$, comparisons $N$
2: **for** $t = 1, 2, \ldots$ **do**
3:     Sample $s_t \sim \mathcal{N}(0, I_d)$
4:     Query oracle $N$ times on $(\theta^t, \theta^t + \alpha_t s_t)$, get $\tilde{R}_{t,1}, \ldots, \tilde{R}_{t,N} \in [-1, 1]$
5:     $S_t = \sum_{n=1}^{N} \tilde{R}_{t,n}$
6:     $\theta^{t+1} = \begin{cases} \theta^t + \alpha_t s_t, & S_t > 0, \\ \theta^t, & \text{otherwise.} \end{cases}$
7: **end for**

---

**Notation and decision events.** Let

$$\begin{cases} \Delta_t := f(\theta^t + \alpha_t s_t) - f(\theta^t), \\ X_t := \{S_t > 0\}, \\ Y_t := \{\Delta_t < 0\}. \end{cases}$$

Since the update is accept–or–stay, we have

$$f(\theta^{t+1}) = f(\theta^t) + \Delta_t \mathbf{1}_{X_t}.$$

To derive a descent inequality we control $\mathbb{E}[\Delta_t \mathbf{1}_{X_t} \mid \theta^t]$ via

$$\Delta_t \mathbf{1}_{X_t} = \underbrace{\Delta_t \mathbf{1}_{Y_t}}_{\text{true-improvement term}} + \underbrace{\Delta_t (\mathbf{1}_{X_t} - \mathbf{1}_{Y_t})}_{\text{ranking-error term}}.$$

The first term coincides with the exact comparator case and can be upper bounded using $L_f$-smoothness and Gaussian symmetry.

---

**Lemma 3.1.** *Assume that $f(x) = g(Ax)$ for some matrix $A \in \mathbb{R}^{k \times d}$ with full row rank $\mathrm{rank}(A) = k \leq d$ and some function $g : \mathbb{R}^k \to \mathbb{R}$, and that $f$ is $L_f$-smooth. Under Algorithm 2, for all $t \geq 1$, we have:*

$$\mathbb{E}\big[\Delta_t \, \mathbf{1}_{Y_t} \,\big|\, \theta^t\big] \;\leq\; -\frac{\alpha_t}{\sqrt{2\pi}} \, \|\nabla f(\theta^t)\|_2 \;+\; \frac{L_f}{2} \, k \, \alpha_t^2.$$

---

The second term captures ranking mistakes. Under Assumption 1.2, consider the queried pair $(x, y) = (\theta^t, \theta^t + \alpha_t s_t)$ and note that $\Delta(x, y) = f(x) - f(y) = -\Delta_t$. A single query then has *aligned* mean at least $\rho(|\Delta_t|)$ and conditional second moment at most $C \rho(|\Delta_t|)$. Repeating the same pair $N$ times and taking the sign of the sum yields an exponentially small error probability with exponent proportional to $N \rho(|\Delta_t|)$.

---

**Lemma 3.2.** *Assume that Assumption 1.2 holds and condition on $(\theta^t, s_t)$ with $\Delta_t \neq 0$. Under Algorithm 2, we have:*

$$\mathbb{E}\big[\big| \, \mathbf{1}_{X_t} - \mathbf{1}_{Y_t} \, \big| \,\big|\, \theta^t, s_t\big] \;\leq\; \exp\left(-\frac{N\rho(|\Delta_t|)}{2C + \frac{4}{3}}\right).$$

---

Lemma 3.2 controls the probability of a wrong aggregate decision on a fixed pair. To translate this into a bound on the *expected* ranking-error term $\mathbb{E}[\Delta_t(\mathbf{1}_{X_t} - \mathbf{1}_{Y_t}) \mid \theta^t]$, we decompose into the "large-gap" regime $|\Delta_t| > r$ and the "small-gap" regime $|\Delta_t| \leq r$, where $\rho(t) \geq ct$ on $[0, r]$. This yields an exponentially small contribution for $|\Delta_t| > r$ and a $1/N$ contribution for $|\Delta_t| \leq r$. Let $\gamma_{N,r} := \exp\left(-\frac{N\rho(r)}{2C + \frac{4}{3}}\right)$.

---

**Lemma 3.3.** *Assume that $f(x) = g(Ax)$ for some matrix $A \in \mathbb{R}^{k \times d}$ with full row rank $\mathrm{rank}(A) = k \leq d$ and some function $g : \mathbb{R}^k \to \mathbb{R}$, and that $f$ is $L_f$-smooth. Define $\mathcal{E}_t := \left|\mathbb{E}\big[\Delta_t(\mathbf{1}_{X_t} - \mathbf{1}_{Y_t}) \mid \theta^t\big]\right|$. Assume that Assumption 1.2 holds. Under Algorithm 2, for all $t \geq 1$, we have:*

$$\mathcal{E}_t \;\leq\; \gamma_{N,r}\left(\alpha_t\sqrt{\frac{2}{\pi}} \, \|\nabla f(\theta^t)\|_2 + \frac{L_f}{2} \, k \, \alpha_t^2\right) + \frac{2C + \frac{4}{3}}{e \, cN}.$$

---

**One-step descent.** Combining the decomposition $\Delta_t \mathbf{1}_{X_t} = \Delta_t \mathbf{1}_{Y_t} + \Delta_t(\mathbf{1}_{X_t} - \mathbf{1}_{Y_t})$ with Lemma 3.1 and Lemma 3.3 yields a descent inequality with an explicit penalty induced by the gap-dependent confidence oracle.

---

**Proposition 3.4.** *Assume that $f(x) = g(Ax)$ for some matrix $A \in \mathbb{R}^{k \times d}$ with full row rank $\mathrm{rank}(A) = k \leq d$ and some function $g : \mathbb{R}^k \to \mathbb{R}$, and that $f$ is $L_f$-smooth. Assume that Assumption 1.2 holds. Under Algorithm 2, for all $t \geq 1$, we have:*

$$\frac{\alpha_t}{\sqrt{2\pi}} \, (1 - \gamma_{N,r}) \, \mathbb{E}\|\nabla f(\theta^t)\|_2 \;\leq\; \mathbb{E}\big[f(\theta^t) - f(\theta^{t+1})\big]$$
$$+ \frac{L_f}{2} \, k \, (1 + \gamma_{N,r}) \, \alpha_t^2 + \frac{2C + \frac{4}{3}}{e \, cN}.$$

---

Assuming moreover that $f$ is bounded below, averaging Proposition 3.4 with a constant stepsize yields the following rate.

---

**Theorem 3.5.** *Assume that $f(x) = g(Ax)$ for some matrix $A \in \mathbb{R}^{k \times d}$ with full row rank $\mathrm{rank}(A) = k \leq d$ and some function $g : \mathbb{R}^k \to \mathbb{R}$, and that $f$ is $L_f$-smooth and bounded below, i.e., $f^\star := \inf_{x \in \mathbb{R}^d} f(x) > -\infty$. Assume that Assumption 1.2 holds. Under Algorithm 2 with a constant stepsize $\alpha_t = \alpha$ and a fixed number of comparisons $N$, for all $T \geq 1$, we have:*

$$\frac{1}{T}\sum_{t=1}^{T} \mathbb{E}\|\nabla f(\theta^t)\|_2 \;\leq\; \frac{\sqrt{2\pi}}{\alpha(1 - \gamma_{N,r})}\left(\frac{\Delta f}{T}\right.$$
$$\left. + \frac{L_f}{2} \, k \, (1 + \gamma_{N,r}) \, \alpha^2 + \frac{2C + \frac{4}{3}}{e \, cN}\right),$$

*where $\Delta f = f(\theta^1) - f^\star$.*

---

*Remark 3.6.* Fix $\varepsilon \in (0, 1)$ and choose $N$ so that: $\gamma_{N,r} \leq \frac{1}{2}$. For instance, it suffices to take $N \geq \frac{\left(2C + \frac{4}{3}\right)\log 2}{\rho(r)}$. Then $1 - \gamma_{N,r} \geq \frac{1}{2}$ and $1 + \gamma_{N,r} \leq \frac{3}{2}$. Applying Theorem 3.5 gives:

$$\frac{1}{T}\sum_{t=1}^{T} \mathbb{E}\|\nabla f(\theta^t)\|_2 \;\leq\; \frac{2\sqrt{2\pi}}{\alpha}\left(\frac{\Delta f}{T} + \frac{3L_f k}{4} \, \alpha^2 + \frac{l_{c,C}}{N}\right),$$

where $l_{c,C} := \frac{2C + \frac{4}{3}}{e \, c}$. By choosing:

$$\begin{cases} \alpha := \dfrac{2\varepsilon}{9\sqrt{2\pi} \, L_f k}, \\[2mm] T \geq \dfrac{54\pi \, L_f k \, \Delta f}{\varepsilon^2}, \\[2mm] N \geq \dfrac{54\pi \, L_f k \, l_{c,C}}{\varepsilon^2}, \end{cases}$$

and additionally enforcing $N \geq \frac{(2C + \frac{4}{3})\log 2}{\rho(r)}$ to ensure $\gamma_{N,r} \leq \frac{1}{2}$, we obtain: $\frac{1}{T}\sum_{t=1}^{T} \mathbb{E}\|\nabla f(\theta^t)\|_2 \leq \varepsilon$. Therefore the total number of pairwise comparisons satisfies:

$$NT \;=\; \mathcal{O}\left(\frac{L_f^2 \, k^2 \, \Delta f}{\varepsilon^4} \cdot \frac{C + 1}{c}\right).$$

*Table 2.* Per-iteration update of NCRS, MeZO, and RSGF

| Method | Iteration $t$ (two minibatch evaluations) |
| --- | --- |
| **NCRS (ours)** | Sample $s_t \sim \mathcal{N}(0, I_d)$. Evaluate $\hat{f}_{\mathcal{B}_t}(\theta^t)$ and $\hat{f}_{\mathcal{B}_t}(\theta^t + \alpha s_t)$ on the *same* minibatch $\mathcal{B}_t$. Set $\widehat{\Delta}_t = \hat{f}_{\mathcal{B}_t}(\theta^t + \alpha s_t) - \hat{f}_{\mathcal{B}_t}(\theta^t)$ and $\widetilde{R}_t = \text{sign}(\widehat{\Delta}_t) \in \{-1, +1\}$. **Update:** if $\widetilde{R}_t = -1$ accept $\theta^{t+1} = \theta^t + \alpha s_t$, else reject $\theta^{t+1} = \theta^t$. |
| **MeZO** | Sample $u_t \sim \mathcal{N}(0, I_d)$. Evaluate $\hat{f}_{\mathcal{B}_t}(\theta^t + \mu u_t)$ and $\hat{f}_{\mathcal{B}_t}(\theta^t - \mu u_t)$ on the *same* minibatch $\mathcal{B}_t$. Form the two-sided estimator $\hat{g}_t = \frac{\hat{f}_{\mathcal{B}_t}(\theta^t + \mu u_t) - \hat{f}_{\mathcal{B}_t}(\theta^t - \mu u_t)}{2\mu} u_t$. **Update:** $\theta^{t+1} = \theta^t - \eta_t \hat{g}_t$. |
| **RSGF** | Sample $u_t \sim \mathcal{N}(0, I_d)$. Evaluate $\hat{f}_{\mathcal{B}_t}(\theta^t + \mu u_t)$ and $\hat{f}_{\mathcal{B}_t}(\theta^t)$ on the *same* minibatch $\mathcal{B}_t$. Form $\hat{g}_t = \frac{\hat{f}_{\mathcal{B}_t}(\theta^t + \mu u_t) - \hat{f}_{\mathcal{B}_t}(\theta^t)}{\mu} u_t$. **Update:** $\theta^{t+1} = \theta^t - \eta_t \hat{g}_t$. |

# 4. Numerical Experiments

## 4.1. Intrinsic Dimension Effect

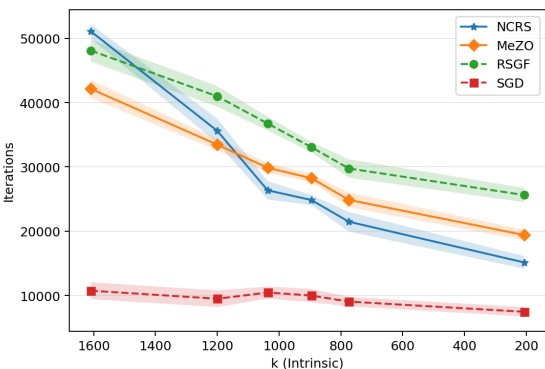

*Figure 1.* **Zeroth-order methods adapt to intrinsic dimension.** Number of iterations required to reach a fixed target accuracy across different language models fine-tuning settings with *decreasing* intrinsic dimension. Shaded areas represent the 95% CI over 5 runs.

We investigate the effect of intrinsic dimension on convergence using medium-sized masked language models. Despite having hundreds of millions of parameters, these models exhibit significantly lower intrinsic dimensions on downstream tasks after pre-training (Aghajanyan et al., 2021; Malladi et al., 2023). We build on the setup in Aghajanyan et al. (2021), which measured intrinsic dimension for various medium-sized models on two fine-tuning tasks (MRPC and QQP from the GLUE benchmark (Wang et al., 2018)), but now perform fine-tuning with zeroth-order methods.

Each fine-tuning task is represented by a training dataset of context-target pairs $\mathcal{D} = \{(x_i, y_i)\}_{i=1,\dots,N}$, where both $x_i$ and $y_i$ are sequences of tokens. Given a pretrained language model that defines $p_\theta(y_t|x, y_{<t})$, the training objective is to

minimize the negative log-likelihood loss:

$$\min_\theta \sum_{i=1}^N \left[ -\sum_{t=1}^{|y_i|} \log p_\theta(y_{i,t}|x_i, y_{i,<t}) \right]$$

Instead of only varying tasks that have different intrinsic dimensions (see Table 3), we vary either the model itself (larger models with more parameters and sufficient pre-training tend to compress downstream datasets better and therefore have lower intrinsic dimensionality) or directly the dataset (easier, more compressible datasets have lower intrinsic dimension). For each setup, we first use standard stochastic gradient descent (SGD) to measure the best accuracy achieved on a validation set and record the number of iterations, i.e., gradient steps, required to reach it. Then, for the same setup, we optimize with NCRS, MeZO (Malladi et al., 2023), and the one-sided finite-difference method RSGF (Ghadimi & Lan, 2013), and record the number of iterations, defined as two function evaluations for zeroth-order methods, needed to exceed the accuracy reached by SGD. For each method, we sweep over the hyperparameters and use the optimal configuration for recording the number of iterations, as discussed in Appendix B.1.

In Figure 1, we find that as the intrinsic dimension decreases, even when the number of parameters increases in some cases, the zeroth-order methods require fewer iterations to reach SGD's accuracy. NCRS, MeZO, and RSGF all exhibit this trend, suggesting that their efficiency is tied more closely to the intrinsic dimension than to the ambient parameter dimension. Among the zeroth-order methods, MeZO provides a strong baseline, while NCRS remains competitive and consistently outperforms RSGF. In the lower intrinsic-dimension regimes, NCRS further closes the gap with SGD, making comparison-based random search an efficient alternative on such tasks. It is also important to note that zeroth-order methods are not only viable for easier fine-tuning tasks; rather, their efficiency appears to be driven primarily by better pre-training and larger models. Thus, scaling does not hurt but instead emphasizes the efficiency of zeroth-order methods.

*Table 3.* Intrinsic dimension $k$ for different sentence understanding tasks and pre-trained models with number of parameters $d$.

| Model | Data set | $d$ | $k$ |
| --- | --- | --- | --- |
| BERT-Base | MRPC | 110M | 1608 |
| BERT-Large | QQP | 340M | 1200 |
| BERT-Large | MRPC | 340M | 1037 |
| RoBERTa-Base | MRPC | 125M | 896 |
| RoBERTa-Large | QQP | 355M | 774 |
| RoBERTa-Large | MRPC | 355M | 207 |

To further isolate the intrinsic-dimension effect, Appendix G reports a controlled synthetic ridge experiment in which the

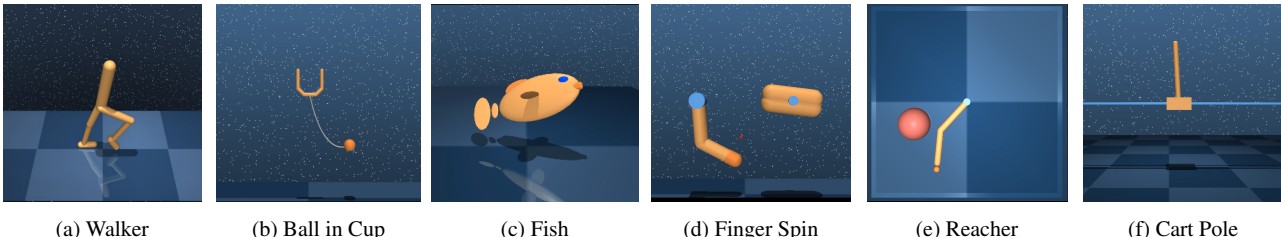

| (a) Walker | (b) Ball in Cup | (c) Fish | (d) Finger Spin | (e) Reacher | (f) Cart Pole |

*Figure 2.* **Environments.** We evaluate preference-based reinforcement learning algorithms on six locomotion tasks.

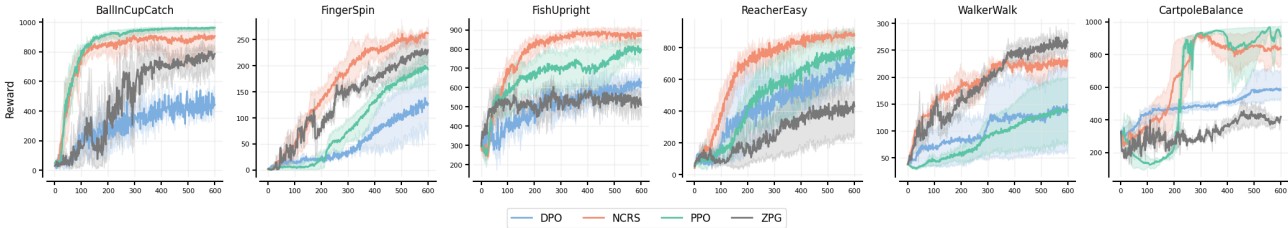

*Figure 3.* **Comparison between preference learning algorithms on DM Control Suite tasks.** Results show the mean ground truth reward and standard error (y-axis) over 600 episodes (x-axis) for 5 random seeds where each algorithm collects 64 trajectories per episode.

ambient dimension $d$ is fixed and the intrinsic dimension $k$ is varied. This experiment exactly matches the ridge model analyzed in the theory and empirically confirms the predicted linear dependence on $k$ for the uniform-margin oracle and quadratic total-comparison dependence for the confidence-oracle setting.

### 4.2. Preference-Based Reinforcement Learning

We consider the reinforcement learning problem where no access to a reward function is given, and instead, we can only receive trajectory preferences from an expert. The goal is then to learn a policy that maximizes the expert's utility function. In addition, we assume that the expert not only generates a binary preference but also a confidence score (Touvron et al., 2023; Kim et al., 2024). NCRS with confidence-weighted vote naturally fits this setting alongside commonly used preference learning algorithms like RLHF (Christiano et al., 2017), iterative DPO (Rafailov et al., 2023; Guo et al., 2024), or zeroth-order alternative ZPG (Zhang & Ying, 2025). RLHF methods fit the utility function by assuming the probabilistic link is logistic, then maximizing the utility with an online RL algorithm like PPO (Schulman et al., 2017). DPO instead bypasses fitting the reward function through a reparameterization of the reward optimization objective. Both algorithms can incorporate the confidence score directly into the objective through a soft label in the cross-entropy loss. On the other hand, ZPG estimates the soft label with a majority vote over $M$ independent human experts.

To evaluate these algorithms, we simulate expert feedback to solve RL tasks in the DeepMind Control Suite (Tassa et al., 2018; Tunyasuvunakool et al., 2020). We use simulated feedback where preferences are based on the environment's task reward function, using the logistic function as a probabilistic link $\sigma$. We incorporate the setup from the B-Pref benchmark (Lee et al., 2021) to account for various human irrationalities (mistakes, myopia, etc.) ensuring a more realistic setup. The confidence score for baselines like RLHF and DPO comes from $\sigma$ directly, whereas for NCRS, we apply the transformation $2\sigma(\Delta) - 1$ to map the probability to the signed confidence score $\tilde{R} \in [-1, 1]$. While the algorithms use different assumptions and compute budgets, we evaluate them under the same query budget, with additional hyperparameter details found in Appendix B.2. In Figure 3, we find NCRS achieves competitive performance against other baselines.

## 5. Limitations and Conclusion

**Limitations.** Our guarantees rely on a fixed ridge-type active-subspace model and smoothness of the objective. This model isolates intrinsic-dimension effects, but practical objectives may only approximately satisfy such a structure or may have active subspaces that vary along the optimization trajectory. The theoretical step sizes and vote counts also depend on problem constants, motivating adaptive variants.

**Conclusion.** We studied smooth nonconvex optimization under noisy pairwise-comparison feedback. Our analysis shows that even when comparisons are unreliable near ties, they can still provide enough directional information to obtain stationarity guarantees. Under low-dimensional structure, NCRS further replaces dependence on the ambient dimension $d$ by dependence on the intrinsic dimension $k$.

## Impact Statement

This work is primarily theoretical and aims to advance optimization from comparison feedback. We do not identify direct negative societal consequences specific to the proposed analysis. However, in applications such as recommendation, ranking, or model alignment, comparison data may be biased or unrepresentative, and deployments should account for this.

## Acknowledgments

The authors gratefully acknowledge the computing resources provided by the Toubkal Supercomputer at UM6P, Morocco (Kissami et al., 2025).

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

## A. Preliminary

**Lemma A.1.** *Let* $f : \mathbb{R}^d \to \mathbb{R}$ *be differentiable and let* $k \in \{1, \dots, d\}$. *The following statements are equivalent:*

1. *(**Gradient lives in a fixed** $k$**-subspace**) There exists a linear subspace* $V \subset \mathbb{R}^d$ *with* $\dim(V) = k$ *such that:*

$$\forall x \in \mathbb{R}^d, \ \nabla f(x) \in V.$$

2. *(**Ridge representation**) There exist a full row-rank matrix* $A \in \mathbb{R}^{k \times d}$ *and a differentiable function* $g : \mathbb{R}^k \to \mathbb{R}$ *such that:*

$$\forall x \in \mathbb{R}^d, \ \ f(x) = g(Ax),$$

*Proof.* **(2)** $\Rightarrow$ **(1)**. Let $x \in \mathbb{R}^d$. We have $\langle \nabla f(x), v \rangle = 0$ for all $v \in Ker(A)$. This implies that $\nabla f(x) \in \text{range}(A^\top) = Ker(A)^\perp$, and we have $\dim(\text{range}(A^\top)) = k$.

**(1)** $\Rightarrow$ **(2)**. Assume there exists a $k$-dimensional subspace $V$ such that $\nabla f(x) \in V$ for all $x \in \mathbb{R}^d$. Using the mean value theorem, we prove that $\forall x \in \mathbb{R}^d, \forall u \in V^\perp, \ f(x + u) = f(x)$. Let $A$ be a matrix such that $\text{range}(A^\top) = V$. This implies that for all $x, y \in \mathbb{R}^d$, if $Ax = Ay$, then we have $f(x) = f(y)$.

Since $A$ has full row rank, the map $x \mapsto Ax$ is surjective onto $\mathbb{R}^k$, so we can define

$$g : \mathbb{R}^k \to \mathbb{R}, \ g(z) := f(x) \ \text{ for any } x \in \mathbb{R}^d \text{ such that } Ax = z.$$

This is well-defined because $Ax = Ay$ implies $f(x) = f(y)$. By construction, for all $x \in \mathbb{R}^d$ we have $f(x) = g(Ax)$. Let $B \in \mathbb{R}^{d \times k}$ be a right inverse of $A$. Then $A(Bz) = z$ for all $z \in \mathbb{R}^k$, hence by the definition of $g$ we have $g(z) = f(Bz)$. Since $f$ is differentiable and $z \mapsto Bz$ is linear, $g$ is differentiable. $\qquad\square$

## B. Hyperparameters

### B.1. LLM Setting

To make sure we use a strong configuration for each method before recording the number of iterations needed to reach the target accuracy, we tune the relevant hyperparameters using validation loss. For SGD and NCRS, we sweep the learning rate $\alpha$, as shown in Figure 4. For RSGF, we sweep both the learning rate $\alpha$ and the smoothing parameter $\mu$, as shown in Figure 5. For MeZO, we follow the same validation-based tuning protocol over its learning rate and perturbation parameter. Each algorithm then uses the configuration that records the lowest validation loss.

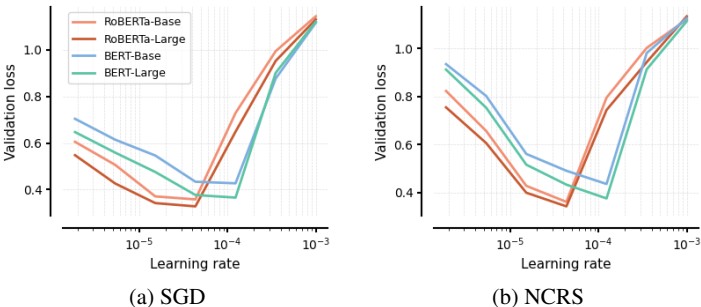

(a) SGD         (b) NCRS

*Figure 4.* **Learning rate sweep.** Validation loss for different models on the MRPC task for SGD and NCRS.

### B.2. Preference-Based RL

Policies are parametrized by a neural network with two hidden layers of 256 units and tanh activations. The network outputs the mean of a Gaussian distribution, while the standard deviation is learned as a separate parameter. We initialize weights using orthogonal initialization and set biases to zero. Complete hyperparameters for all algorithms are listed in Table 4.

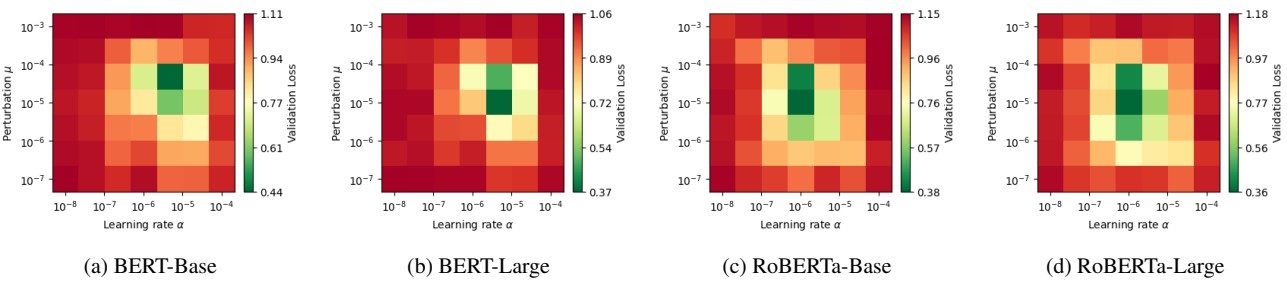

(a) BERT-Base  (b) BERT-Large  (c) RoBERTa-Base  (d) RoBERTa-Large

*Figure 5.* **2D parameter sweep.** Validation loss for different models on the MRPC task for RSGF with varying learning rate $\alpha$ and perturbation $\mu$.

## C. Convergence Analysis of NCRS Algorithm

**Lemma C.1** (Lemma 2.1). *Assume that $f$ admits a ridge representation $f(x) = g(Ax)$ for some matrix $A \in \mathbb{R}^{k \times d}$ with full row rank $\operatorname{rank}(A) = k \leq d$ and some function $g : \mathbb{R}^k \to \mathbb{R}$, and that $f$ is $L_f$-smooth. Let $(\theta^t)$ be the iterates produced by Algorithm 1, and suppose the ranking oracle satisfies Assumption 1.1 with parameter $p$. Then for all $t \geq 1$, we have:*

$$p\alpha_t \sqrt{\frac{2}{\pi}} \mathbb{E}\|\nabla f(\theta^t)\|_2 \leq \mathbb{E}[f(\theta^t) - f(\theta^{t+1})] + \frac{L_f k \alpha_t^2}{2}.$$

*Proof of Lemma 2.1.* Fix $t \geq 1$ and condition on $\theta^t$. Let $\mathcal{A}_t := \left\{ R(\theta^t, \theta^t + \alpha_t s_t) = +1 \right\}$ be the event that NCRS accepts the candidate $\theta^t + \alpha_t s_t$. We remark that:

$$f(\theta^t) - f(\theta^{t+1}) = \left( f(\theta^t) - f(\theta^t + \alpha_t s_t) \right) \mathbf{1}_{\mathcal{A}_t},$$

and therefore:

$$\mathbb{E}\left[ f(\theta^t) - f(\theta^{t+1}) \mid \theta^t, s_t \right] = \left( f(\theta^t) - f(\theta^t + \alpha_t s_t) \right) \mathbb{P}\left( \mathcal{A}_t \mid \theta^t, s_t \right).$$

By Assumption 1.1, if $f(\theta^t + \alpha_t s_t) < f(\theta^t)$ then $\mathbb{P}(\mathcal{A}_t \mid \theta^t, s_t) \geq \frac{1}{2} + p$, while if $f(\theta^t + \alpha_t s_t) > f(\theta^t)$ then $\mathbb{P}(\mathcal{A}_t \mid \theta^t, s_t) \leq \frac{1}{2} - p$. This implies that:

$$\left( f(\theta^t) - f(\theta^t + \alpha_t s_t) \right) \mathbb{P}\left( \mathcal{A}_t \mid \theta^t, s_t \right) \geq \left( \tfrac{1}{2} - p \right)\left( f(\theta^t) - f(\theta^t + \alpha_t s_t) \right) + 2p \left[ f(\theta^t) - f(\theta^t + \alpha_t s_t) \right]_+,$$

where $[u]_+ := \max\{u, 0\}$. It follows that:

$$\mathbb{E}\left[ f(\theta^t) - f(\theta^{t+1}) \mid \theta^t \right] \geq \left( \tfrac{1}{2} - p \right) \mathbb{E}\left[ f(\theta^t) - f(\theta^t + \alpha_t s_t) \mid \theta^t \right] + 2p\, \mathbb{E}\left[ \left[ f(\theta^t) - f(\theta^t + \alpha_t s_t) \right]_+ \mid \theta^t \right]. \quad (1)$$

Let $P := A^\top (AA^\top)^{-1} A$ the orthogonal projector onto $\operatorname{range}(A^\top)$. Since $f$ is $L_f$ smooth, we have:

$$
\begin{aligned}
f(\theta^t + \alpha_t s_t) &= g\left( A\theta^t + \alpha_t A s_t \right) \\
&= g(A\theta^t + \alpha_t A P s_t) \\
&= f(\theta^t + \alpha_t P s_t) \\
&\leq f(\theta^t) + \alpha_t \left\langle \nabla f(\theta^t), P s_t \right\rangle + \frac{L_f}{2} \alpha_t^2 \|P s_t\|_2^2 \\
&= f(\theta^t) + \alpha_t \left\langle P^\top \nabla f(\theta^t), s_t \right\rangle + \frac{L_f}{2} \alpha_t^2 \|P s_t\|_2^2 \\
&= f(\theta^t) + \alpha_t \left\langle P \nabla f(\theta^t), s_t \right\rangle + \frac{L_f}{2} \alpha_t^2 \|P s_t\|_2^2.
\end{aligned}
$$

We used in the last inequality that $P^\top = P$, since $P$ is an orthogonal projector.

*Table 4.* **Hyperparameters for Preference-Based RL Algorithms (PPO, DPO, NCRS, ZPG).**

| Hyperparameter | Value |
| --- | --- |
| Number of trajectories collected per episode (**for all algorithms**) | 64 |
| **Proximal Policy Optimization (PPO)** | |
| Actor Learning Rate Schedule | Warmup-Stable |
| Actor Warmup Steps | 20 |
| Actor Maximum Learning Rate | $3 \times 10^{-4}$ |
| Critic MLP dimensions | $(256, 256)$ |
| Critic Learning Rate | $1 \times 10^{-3}$ |
| Clip Parameter $\epsilon$ | 0.2 |
| GAE $\lambda$ | 0.95 |
| Discount Factor $\gamma$ | 0.99 |
| Reward Model MLP dimensions | $(256, 256)$ |
| Reward Model Batch Size | 64 |
| Reward Model Learning Rate | $4 \times 10^{-3}$ |
| Optimizer | Adam (Kingma & Ba, 2015) |
| Number of gradient steps per update (same for actor, critic, and reward model) | 10 |
| Maximum gradient norm | 0.5 |
| **Direct Preference Optimization (DPO)** | |
| $\beta$ | 0.1 |
| Learning Rate | $3 \times 10^{-4}$ |
| Batch Size | 32 |
| Number of gradient steps per update | 5 |
| Optimizer | Adam (Kingma & Ba, 2015) |
| Maximum gradient norm | 0.5 |
| **Noisy Pairwise-Comparison Random Search (NCRS)** | |
| Learning Rate Schedule | Cosine-Decay |
| Number of decaying steps | 480 |
| Maximum Learning Rate | $4 \times 10^{-2}$ |
| Minimum Learning Rate | $4 \times 10^{-3}$ |
| **Zeroth-order Policy Gradient from Human Feedback (ZPG)** | |
| Smoothing parameter $\mu$ | $4.5 \times 10^{-2}$ |
| Learning Rate | $2 \times 10^{-3}$ |

Note that:

$$\forall x \in \mathbb{R}^d, \ \nabla f(x) \ \in \ \text{range}(A^\top). \tag{2}$$

Indeed, for any $v \in \ker(A)$, the map $t \mapsto f(x + tv)$ is constant on $\mathbb{R}$, hence $\langle \nabla f(x), v \rangle = 0$ for all $v \in \ker(A)$. Therefore $\nabla f(x) \perp \ker(A)$, which implies $\nabla f(x) \in \text{range}(A^\top)$ and thus $P\nabla f(x) = \nabla f(x)$ for all $x \in \mathbb{R}^d$. Consequently:

$$f(\theta^t) - f(\theta^t + \alpha_t s_t) \geq -\alpha_t \left\langle \nabla f(\theta^t), s_t \right\rangle - \frac{L_f}{2} \alpha_t^2 \left\| P s_t \right\|_2^2.$$

This implies that:

$$\mathbb{E}\left[ f(\theta^t) - f(\theta^t + \alpha_t s_t) \mid \theta^t \right] \geq -\frac{L_f}{2} \alpha_t^2 \mathbb{E} \| P s_t \|_2^2. \tag{3}$$

It also implies that:

$$
\begin{aligned}
\left[f(\theta^t) - f(\theta^t + \alpha_t s_t)\right]_+ &\geq \left[-\alpha_t \left\langle \nabla f(\theta^t), s_t \right\rangle - \frac{L_f}{2}\alpha_t^2 \|Ps_t\|_2^2\right]_+ \\
&\geq \alpha_t \left[-\left\langle \nabla f(\theta^t), s_t \right\rangle\right]_+ - \frac{L_f}{2}\alpha_t^2 \|Ps_t\|_2^2 \\
&= \alpha_t \frac{-\left\langle \nabla f(\theta^t), s_t \right\rangle + |\left\langle \nabla f(\theta^t), s_t \right\rangle|}{2} - \frac{L_f}{2}\alpha_t^2 \|Ps_t\|_2^2
\end{aligned}
$$

It follows that:

$$
\mathbb{E}\left[\left[f(\theta^t) - f(\theta^t + \alpha_t s_t)\right]_+ \Big| \theta^t\right] \geq \frac{\alpha_t}{2}\mathbb{E}\left[|\left\langle \nabla f(\theta^t), s_t \right\rangle| \, \Big| \, \theta^t\right] - \frac{L_f}{2}\alpha_t^2 \mathbb{E}\left[\|Ps_t\|_2^2\right]. \tag{4}
$$

Combining the inequalities 1, 3 and 4, we obtain:

$$
\mathbb{E}\left[f(\theta^t) - f(\theta^{t+1}) \mid \theta^t\right] \geq -(\frac{1}{2} - p)\frac{L_f \alpha_t^2}{2}\mathbb{E}\|Ps_t\|_2^2 + p\alpha_t \mathbb{E}\left[|\left\langle \nabla f(\theta^t), s_t \right\rangle| \, \big| \, \theta^t\right] - pL_f \alpha_t^2 \mathbb{E}\|Ps_t\|_2^2.
$$

We remark that $\mathbb{E}\left[|\left\langle \nabla f(\theta^t), s_t \right\rangle| \, \big| \, \theta^t\right] = \|\nabla f(\theta^t)\|_2 \, \mathbb{E}_{u\sim\mathcal{N}(0,1)}[|u|] = \sqrt{\frac{2}{\pi}} \|\nabla f(\theta^t)\|_2$. Therefore:

$$
p\alpha_t \sqrt{\frac{2}{\pi}}\mathbb{E}\|\nabla f(\theta^t)\|_2 \leq \mathbb{E}[f(\theta^t) - f(\theta^{t+1})] + \frac{L_f \alpha_t^2}{2}\mathbb{E}\|Ps_t\|_2^2.
$$

We also note that, for $s \sim \mathcal{N}(0, I_d)$, we have:

$$
\mathbb{E}\|Ps\|_2^2 = \mathbb{E}\left[s^\top P^\top P s\right] = \mathrm{tr}\left(P^\top P \, \mathbb{E}[ss^\top]\right) = \mathrm{tr}(P^\top P) = \mathrm{tr}(P^2) = \mathrm{tr}(P) = k,
$$

where we used $\mathbb{E}[ss^\top] = I_d$, $P^\top = P$, $P^2 = P$, and $\mathrm{tr}(P) = \mathrm{rank}(P) = k$. We deduce that:

$$
p\alpha_t \sqrt{\frac{2}{\pi}}\mathbb{E}\|\nabla f(\theta^t)\|_2 \leq \mathbb{E}[f(\theta^t) - f(\theta^{t+1})] + \frac{L_f k \alpha_t^2}{2}.
$$

$\square$

## D. Gap-Dependent Confidence Oracle: NCRS Analysis with Confidence-Weighted Vote

**Lemma D.1** (Theorem 3.1). *Assume that $f(x) = g(Ax)$ for some matrix $A \in \mathbb{R}^{k\times d}$ with full row rank $\mathrm{rank}(A) = k \leq d$ and some function $g : \mathbb{R}^k \to \mathbb{R}$, and that $f$ is $L_f$-smooth. Under Algorithm 2, for all $t \geq 1$, we have:*

$$
\mathbb{E}\left[\Delta_t \, \mathbf{1}_{Y_t} \, \big| \, \theta^t\right] \leq -\frac{\alpha_t}{\sqrt{2\pi}}\|\nabla f(\theta^t)\|_2 + \frac{L_f}{2}k\alpha_t^2.
$$

*Proof of Theorem 3.1.* Let $P := A^\top (AA^\top)^{-1}A$ the orthogonal projector onto $\mathrm{range}(A^\top)$. By $L_f$-smoothness, we have:

$$
\begin{aligned}
\Delta_t &= f(\theta^t + \alpha P s_t) - f(\theta^t) \\
&\leq \alpha_t \left\langle \nabla f(\theta^t), Ps_t \right\rangle + \frac{L_f}{2}\alpha_t^2 \|Ps_t\|^2 \\
&= \alpha_t \left\langle \nabla f(\theta^t), s_t \right\rangle + \frac{L_f}{2}\alpha_t^2 \|Ps_t\|^2.
\end{aligned}
$$

Therefore:

$$
\Delta_t \mathbf{1}_{Y_t} \leq \min(\Delta_t, 0) \leq \min(\alpha_t \langle \nabla f(\theta^t), s_t \rangle, 0) + \frac{L_f}{2}\alpha_t^2 \|Ps_t\|_2^2,
$$

where we used $\min(y+b, 0) \le \min(y, 0) + b$ for any $b \ge 0$. Taking conditional expectation and using $s_t \sim \mathcal{N}(0, I_d)$,

$$
\begin{aligned}
\mathbb{E}\big[\Delta_t \, \mathbf{1}_{B_t} \,\big|\, \theta^t\big] &\le \alpha_t \, \mathbb{E}\big[\min\{\langle \nabla f(\theta^t), s_t \rangle, 0\} \,\big|\, \theta^t\big] + \tfrac{L_f}{2}\alpha_t^2 \, \mathbb{E}\big[\|Ps_t\|_2^2\big] \\
&= \alpha_t \, \mathbb{E}\left[\left. \frac{\langle \nabla f(\theta^t), s_t \rangle - |\langle \nabla f(\theta^t), s_t \rangle|}{2} \,\right|\, \theta^t\right] + \tfrac{L_f}{2}\, k \, \alpha_t^2 \\
&= -\frac{\alpha_t}{2} \, \mathbb{E}\big[|\langle \nabla f(\theta^t), s_t \rangle| \,\big|\, \theta^t\big] + \tfrac{L_f}{2}\, k \, \alpha_t^2 \\
&= \frac{-\alpha_t}{\sqrt{2\pi}} \, \big\| \nabla f\left(\theta^t\right) \big\|_2 + \tfrac{L_f}{2}\, k \, \alpha_t^2.
\end{aligned}
$$

$\square$

---

**Lemma D.2.** *Assume that* $\Delta_t = f(\theta^t + \alpha_t s_t) - f(\theta^t) \ne 0$. *Let* $\tilde{R}_{t,1}, \ldots, \tilde{R}_{t,N} \in [-1, 1]$ *be the* $N$ *i.i.d. oracle outcomes of Algorithm 2 queried on* $(\theta^t, \theta^t + \alpha_t s_t)$. *Define:*

$$
X_t := \Big\{ \sum_{n=1}^N \tilde{R}_{t,n} > 0 \Big\}, \quad \text{and} \quad Y_t := \{ f(\theta^t) > f(\theta^t + \alpha_t s_t) \}.
$$

*Then*

$$
\mathbb{E}\big[\big|\, \mathbf{1}_{X_t} - \mathbf{1}_{Y_t} \,\big|\,\big|\, \theta^t, s_t\big] \;\le\; \Pr\left(\left. \sum_{n=1}^N \mathrm{sign}\big(f(\theta^t) - f(\theta^t + \alpha_t s_t)\big)\, \tilde{R}_{t,n} \;\le\; 0 \,\right|\, \theta^t, s_t\right).
$$

---

*Proof.* Condition on $(\theta^t, s_t)$ and assume $\Delta_t \ne 0$. We show that whenever $\mathbf{1}_{X_t} \ne \mathbf{1}_{Y_t}$, one must have

$$
\sum_{n=1}^N \mathrm{sign}\big(f(\theta^t) - f(\theta^t + \alpha_t s_t)\big)\, \tilde{R}_{t,n} \;\le\; 0.
$$

There are two cases.

*Case 1:* $f(\theta^t) > f(\theta^t + \alpha_t s_t)$. Then $\mathrm{sign}(f(\theta^t) - f(\theta^t + \alpha_t s_t)) = +1$ and an error $\mathbf{1}_{X_t} \ne \mathbf{1}_{Y_t}$ can only happen if the algorithm rejects, i.e., $\sum_{n=1}^N \tilde{R}_{t,n} \le 0$, which implies the desired inequality.

*Case 2:* $f(\theta^t) < f(\theta^t + \alpha_t s_t)$. Then $\mathrm{sign}(f(\theta^t) - f(\theta^t + \alpha_t s_t)) = -1$ and an error can only happen if the algorithm accepts, i.e., $\sum_{n=1}^N \tilde{R}_{t,n} > 0$. Multiplying by $-1$ yields:

$$
\sum_{n=1}^N \mathrm{sign}\big(f(\theta^t) - f(\theta^t + \alpha_t s_t)\big)\, \tilde{R}_{t,n} = -\sum_{n=1}^N \tilde{R}_{t,n} \le 0.
$$

Thus, in all cases, we have:

$$
\big|\, \mathbf{1}_{X_t} - \mathbf{1}_{Y_t} \,\big| \;\le\; \mathbf{1}_{\left\{ \sum_{n=1}^N \mathrm{sign}\big(f(\theta^t) - f(\theta^t + \alpha_t s_t)\big)\, \tilde{R}_{t,n} \le 0 \right\}}.
$$

By taking conditional expectations given $(\theta^t, s_t)$, we obtain the desired result. $\square$

---

**Lemma D.3** (Bernstein inequality). *Let* $Z_1, \ldots, Z_N$ *be independent random variables such that* $\mathbb{E}[Z_n] = 0$ *and* $|Z_n| \le b$ *almost surely for all* $n$. *Let* $V := \sum_{n=1}^N \mathrm{Var}(Z_n)$. *Then for all* $t \ge 0$, *we have:*

$$
\Pr\left(\sum_{n=1}^N Z_n \le -t\right) \;\le\; \exp\left(-\frac{t^2}{2V + \tfrac{2}{3}bt}\right) \quad \text{and} \quad \Pr\left(\sum_{n=1}^N Z_n \ge t\right) \;\le\; \exp\left(-\frac{t^2}{2V + \tfrac{2}{3}bt}\right).
$$

**Lemma D.4** (Theorem 3.2). *Assume that Assumption 1.2 holds and condition on $(\theta^t, s_t)$ with $\Delta_t \neq 0$. Under Algorithm 2, we have:*

$$\mathbb{E}\big[\big|\, \mathbf{1}_{X_t} - \mathbf{1}_{Y_t}\,\big|\,\big|\,\theta^t, s_t\big] \;\leq\; \exp\left(-\frac{N\rho(|\Delta_t|)}{2C + \frac{4}{3}}\right).$$

*Proof.* Define the "aligned" random variables:

$$Z_{t,n} \;:=\; \operatorname{sign}\big(f(\theta^t) - f(\theta^t + \alpha_t s_t)\big)\, \tilde{R}_{t,n},$$

and let $\mu := \mathbb{E}[Z_{t,1} \mid \theta^t, s_t]$. By Assumption 1.2, we have:

$$\mu \;\geq\; \rho(|\Delta_t|) \quad \text{and} \quad \mathbb{E}[Z_{t,1}^2 \mid \theta^t, s_t] \;=\; \mathbb{E}[\tilde{R}(\theta^t, \theta^t + \alpha_t s_t)^2 \mid \theta^t, s_t] \;\leq\; C\,\rho(|\Delta_t|).$$

Moreover, since $Z_{t,n} \in [-1, 1]$ we have $|Z_{t,n} - \mu| \leq 2$.

By Theorem D.2, we have:

$$\mathbb{E}\big[\big|\, \mathbf{1}_{X_t} - \mathbf{1}_{Y_t}\,\big|\,\big|\,\theta^t, s_t\big] \leq \Pr\left(\sum_{n=1}^{N} Z_{t,n} \leq 0 \,\middle|\, \theta^t, s_t\right)$$

$$= \Pr\left(\sum_{n=1}^{N}(Z_{t,n} - \mu) \leq -N\mu \,\middle|\, \theta^t, s_t\right).$$

Using conditional independence of the $Z_{t,n}$ given $(\theta^t, \theta^t + \alpha_t s_t)$ and Bernstein's inequality for bounded variables $Z_{t,n} - \mu$, we obtain:

$$\Pr\left(\sum_{n=1}^{N}(Z_{t,n} - \mu) \leq -N\mu \,\middle|\, \theta^t, s_t\right) \leq \exp\left(-\frac{(N\mu)^2}{2N\,\mathbb{E}[(Z_{t,1} - \mu)^2 \mid \theta^t, s_t] + \frac{2}{3} \cdot 2 \cdot N\mu}\right).$$

Since $\mathbb{E}[(Z_{t,1} - \mu)^2 \mid \theta^t, s_t] \leq \mathbb{E}[Z_{t,1}^2 \mid \theta^t, s_t] \leq C\rho(|\Delta_t|)$, we obtain:

$$\mathbb{E}\big[\big|\, \mathbf{1}_{X_t} - \mathbf{1}_{Y_t}\,\big|\,\big|\,\theta^t, s_t\big] \leq \exp\left(-\frac{N\mu^2}{2C\rho(|\Delta_t|) + \frac{4}{3}\mu}\right).$$

By denoting $\rho_t := \rho(|\Delta_t|)$. We have: $\mathbb{E}\big[\big|\, \mathbf{1}_{X_t} - \mathbf{1}_{Y_t}\,\big|\,\big|\,\theta^t, s_t\big] \leq \exp\left(-\frac{N\mu^2}{2C\,\rho_t + \frac{4}{3}\mu}\right)$. Consider the function $\phi(s) := \frac{s^2}{2C\,\rho_t + \frac{4}{3}s}$ for $s > 0$. A direct derivative computation shows that $\phi$ is increasing on $(0, \infty)$:

$$\forall s > 0, \;\; \phi'(s) = \frac{2(2C\rho_t)s + \frac{4}{3}s^2}{\left(2C\rho_t + \frac{4}{3}s\right)^2} \;>\; 0.$$

Since $\mu \geq \rho_t$, it follows that:

$$\frac{\mu^2}{2C\,\rho_t + \frac{4}{3}\mu} \;\geq\; \frac{\rho_t^2}{2C\,\rho_t + \frac{4}{3}\,\rho_t} \;=\; \rho_t\,\frac{1}{2C + \frac{4}{3}}.$$

Therefore:

$$\mathbb{E}\big[\big|\, \mathbf{1}_{X_t} - \mathbf{1}_{Y_t}\,\big|\,\big|\,\theta^t, s_t\big] \leq \exp\left(-\frac{N\,\rho(|\Delta_t|)}{2C + \frac{4}{3}}\right).$$

$\square$

**Lemma D.5** (Theorem 3.3). *Assume that $f(x) = g(Ax)$ for some matrix $A \in \mathbb{R}^{k \times d}$ with full row rank $\operatorname{rank}(A) = k \leq d$ and some function $g : \mathbb{R}^k \to \mathbb{R}$, and that $f$ is $L_f$-smooth. Assume that Assumption 1.2 holds. Under Algorithm 2, for all $t \geq 1$, we have:*

$$\left| \mathbb{E}[\Delta_t(\mathbf{1}_{X_t} - \mathbf{1}_{Y_t}) \mid \theta^t] \right| \leq \gamma_{N,r} \left( \alpha_t \sqrt{\frac{2}{\pi}} \|\nabla f(\theta^t)\|_2 + \frac{L_f}{2} k \alpha_t^2 \right) + \frac{2C + \frac{4}{3}}{e \, cN},$$

*where $\gamma_{N,r} := \exp\left( -\frac{N\rho(r)}{2C + \frac{4}{3}} \right)$.*

*Proof of Theorem 3.3.* By Theorem 3.2, for $\Delta_t \neq 0$, we have:

$$\mathbb{E}[ |\mathbf{1}_{X_t} - \mathbf{1}_{Y_t}| \mid \theta^t, s_t ] \leq e^{-\frac{N\rho(|\Delta_t|)}{2C + \frac{4}{3}}},$$

hence, we have:

$$\left| \mathbb{E}[\Delta_t(\mathbf{1}_{X_t} - \mathbf{1}_{Y_t}) \mid \theta^t] \right| \leq \mathbb{E}\left[ |\Delta_t| e^{-\frac{N\rho(|\Delta_t|)}{2C + \frac{4}{3}}} \mid \theta^t \right].$$

We split the expectation into the two regimes $|\Delta_t| > r$ and $|\Delta_t| \leq r$.

*Large-gap regime $|\Delta_t| > r$.* Since $\rho$ is nondecreasing, $\rho(|\Delta_t|) \geq \rho(r) = m_r$, and thus $e^{-\frac{N\rho(|\Delta_t|)}{2C + \frac{4}{3}}} \leq e^{-\frac{N\rho(r)}{2C + \frac{4}{3}}}$. Therefore,

$$\mathbb{E}\left[ |\Delta_t| e^{-N\kappa_{p,C}\rho(|\Delta_t|)} \mathbf{1}_{\{|\Delta_t| > r\}} \mid \theta^t \right] \leq e^{-\frac{N\rho(r)}{2C + \frac{4}{3}}} \mathbb{E}[|\Delta_t| \mid \theta^t].$$

Moreover, using the ridge structure $f(x) = g(Ax)$ with projector $P := A^\top (AA^\top)^{-1} A$, we have $f(\theta^t + \alpha_t s_t) = f(\theta^t + \alpha_t P s_t)$ and $\nabla f(\theta^t) \in \operatorname{range}(A^\top)$, hence, by $L_f$-smoothness of $f$, we have:

$$|\Delta_t| = \left| f(\theta^t + \alpha_t P s_t) - f(\theta^t) \right| \leq \alpha_t \left| \langle \nabla f(\theta^t), s_t \rangle \right| + \frac{L_f}{2} \alpha_t^2 \|P s_t\|_2^2.$$

It follows that:

$$\mathbb{E}[|\Delta_t| \mid \theta^t] \leq \alpha_t \sqrt{\frac{2}{\pi}} \|\nabla f(\theta^t)\|_2 + \frac{L_f}{2} \alpha_t^2 \, \mathbb{E}\|P s_t\|_2^2 = \alpha_t \sqrt{\frac{2}{\pi}} \|\nabla f(\theta^t)\|_2 + \frac{L_f}{2} k \alpha_t^2.$$

*Small-gap regime $|\Delta_t| \leq r$.* On this event we have $\rho(|\Delta_t|) \geq c \, |\Delta_t|$, hence:

$$|\Delta_t| e^{-\frac{N\rho(|\Delta_t|)}{2C + \frac{4}{3}}} \leq |\Delta_t| e^{-\frac{c N |\Delta_t|}{2C + \frac{4}{3}}}.$$

We can verify that $\sup_{x \geq 0} x e^{-ax} = \frac{1}{ea}$ (attained at $x = \frac{1}{a}$), with $a := \frac{cN}{2C + \frac{4}{3}}$, we obtain:

$$|\Delta_t| e^{-\frac{c N |\Delta_t|}{2C + \frac{4}{3}}} \leq \frac{2C + \frac{4}{3}}{e \, cN},$$

and therefore $\mathbb{E}\left[ |\Delta_t| e^{-\frac{c N |\Delta_t|}{2C + \frac{4}{3}}} \mathbf{1}_{\{|\Delta_t| \leq r\}} \mid \theta^t \right] \leq \frac{2C + \frac{4}{3}}{e \, cN}$. Combining the two regimes, we obtain:

$$\left| \mathbb{E}[\Delta_t(\mathbf{1}_{X_t} - \mathbf{1}_{Y_t}) \mid \theta^t] \right| \leq \gamma_{N,r} \left( \alpha_t \sqrt{\frac{2}{\pi}} \|\nabla f(\theta^t)\|_2 + \frac{L_f}{2} k \alpha_t^2 \right) + \frac{2C + \frac{4}{3}}{e \, cN},$$

where $\gamma_{N,r} := e^{-\frac{N\rho(r)}{2C + \frac{4}{3}}}$. $\square$

**Proposition D.6** (Theorem 3.4). *Assume that $f(x) = g(Ax)$ for some matrix $A \in \mathbb{R}^{k \times d}$ with full row rank $\mathrm{rank}(A) = k \leq d$ and some function $g : \mathbb{R}^k \to \mathbb{R}$, and that $f$ is $L_f$-smooth. Assume that Assumption 1.2 holds. Under Algorithm 2, for all $t \geq 1$, we have:*

$$\mathbb{E}\big[f(\theta^{t+1}) \,\big|\, \theta^t\big] \;\leq\; f(\theta^t) - \frac{\alpha_t}{\sqrt{2\pi}} \, (1 - \gamma_{N,r}) \, \|\nabla f(\theta^t)\|_2 + \frac{L_f}{2} \, k \, (1 + \gamma_{N,r}) \, \alpha_t^2 + \frac{2C + \frac{4}{3}}{e \, cN},$$

*where $\gamma_{N,r} := e^{-\frac{N\rho(r)}{2C + \frac{4}{3}}}$.*

*Proof of Theorem 3.4.* Condition on $\theta^t$. By the accept–or–stay update rule,

$$f(\theta^{t+1}) \;=\; f(\theta^t) + \Delta_t \, \mathbf{1}_{X_t}, \text{ where } \Delta_t := f(\theta^t + \alpha_t s_t) - f(\theta^t).$$

Then, we obtain:

$$\mathbb{E}\big[f(\theta^{t+1}) \,\big|\, \theta^t\big] = f(\theta^t) + \mathbb{E}\big[\Delta_t \mathbf{1}_{Y_t} \,\big|\, \theta^t\big] + \mathbb{E}\big[\Delta_t (\mathbf{1}_{X_t} - \mathbf{1}_{Y_t}) \,\big|\, \theta^t\big] .$$

We bound the two terms separately.

**True-improvement term.** By Lemma 3.1, we have:

$$\mathbb{E}\big[\Delta_t \mathbf{1}_{Y_t} \,\big|\, \theta^t\big] \;\leq\; -\frac{\alpha_t}{\sqrt{2\pi}} \, \|\nabla f(\theta^t)\|_2 \;+\; \frac{L_f}{2} \, k \, \alpha_t^2.$$

**Ranking-error term.** By Lemma 3.3, we have:

$$\mathbb{E}\big[\Delta_t (\mathbf{1}_{A_t} - \mathbf{1}_{B_t}) \,\big|\, \theta^t\big] \;\leq\; \gamma_{N,r} \left( \alpha_t \sqrt{\frac{2}{\pi}} \, \|\nabla f(\theta^t)\|_2 + \frac{L_f}{2} \, k \, \alpha_t^2 \right) + \frac{2C + \frac{4}{3}}{e \, cN}.$$

It follows that:

$$\mathbb{E}\big[f(\theta^{t+1}) \,\big|\, \theta^t\big] \;\leq\; f(\theta^t) - \frac{\alpha_t}{\sqrt{2\pi}} \|\nabla f(\theta^t)\|_2 + \frac{L_f}{2} \, k \, \alpha_t^2 + \gamma_{N\,r} \left( \alpha_t \sqrt{\frac{2}{\pi}} \|\nabla f(\theta^t)\|_2 + \frac{L_f}{2} \, k \, \alpha_t^2 \right) + \frac{2C + \frac{4}{3}}{e \, cN}$$

Therefore:

$$\mathbb{E}\big[f(\theta^{t+1})\big] \;\leq\; \mathbb{E}\big[f(\theta^t)\big] - \frac{\alpha_t}{\sqrt{2\pi}} \, (1 - \gamma_{N,r}) \, \mathbb{E}\|\nabla f(\theta^t)\|_2 + \frac{L_f}{2} \, k \, (1 + \gamma_{N,r}) \, \alpha_t^2 + \frac{2C + \frac{4}{3}}{e \, cN}.$$

$\square$

# E. Convergence Analysis of Two-Point Gradient Estimation Method for Ridge Objectives $f(x) = g(Ax)$

## E.1. Results

**Setting.** Throughout this section we keep the same ridge structure $f(x) = g(Ax)$, with $A \in \mathbb{R}^{k \times d}$ and $\mathrm{rank}(A) = k \leq d$, and assume that $f$ is $L_f$-smooth on $\mathbb{R}^d$.

**Two-point gradient estimation method.** As a standard baseline for gradient estimation methods in zeroth-order optimization, we consider the classical RSGF (Ghadimi & Lan, 2013) algorithm, which constructs a stochastic estimate of the gradient. For completeness, we recall the procedure in Algorithm 3.

---

**Algorithm 3** Two-point gradient estimation method

---

1: **Input:** initial point $\theta^1 \in \mathbb{R}^d$, smoothing radius $\mu > 0$, stepsize $\alpha > 0$
2: **for** $t = 1, 2, \ldots$ **do**
3:     Sample a random direction $s_t \sim \mathcal{N}(0, I_d)$
4:     Update:

$$\theta^{t+1} = \theta^t - \alpha \frac{h(\theta^t + \mu s_t) - h(\theta^t)}{\mu} \, s_t$$

5: **end for**

---

**Two-point gradient estimation method automatically adapts to the intrinsic subspace.** In the ridge model $h(x) = g(Ax)$ with $P := A^\top (AA^\top)^{-1} A$, we have $h(x + \eta s) = h(x + \eta P s)$ for all $x, s \in \mathbb{R}^d$ and $\eta > 0$. Hence the two-point difference depends only on the projected direction:

$$\delta_t := \frac{h(\theta^t + \mu s_t) - h(\theta^t)}{\mu} = \frac{h(\theta^t + \mu P s_t) - h(\theta^t)}{\mu}.$$

Although Algorithm 3 updates with $s_t$,

$$\theta^{t+1} = \theta^t - \alpha \, \delta_t \, s_t = \theta^t - \alpha \, \delta_t \, P s_t - \alpha \, \delta_t \, (I - P) s_t,$$

the last term lies in $\ker(A)$ and is invisible to $h$ since $A(I - P) = 0$. Therefore $h(\theta^{t+1}) = h(\theta^t - \alpha \, \delta_t \, P s_t)$.

Equivalently, for optimizing $f$ we may replace $s_t$ by the effective direction $u_t := P s_t \in \text{range}(A^\top)$. The method therefore behaves as a two-point gradient approximation scheme restricted to the $k$-dimensional active subspace, with $u_t \sim \mathcal{N}(0, P)$. Since $P$ has rank $k$, the Gaussian moments of $u_t$ that appear in the analysis scale with $k$. Moreover, the ridge structure implies $\nabla h(x) \in \text{range}(A^\top)$ and thus $P \nabla h(x) = \nabla h(x)$. Together, these facts explain why the smoothness and variance terms in the descent inequality depend on $k$.

Similarly to Lemma 2.1, we establish an analogous one-step descent inequality tailored to Algorithm 3.

---

**Lemma E.1.** *Assume the observed objective is of the form $h(x) = g(Ax)$ for some matrix $A \in \mathbb{R}^{k \times d}$ with full row rank $\text{rank}(A) = k \le d$, and some function $g : \mathbb{R}^k \to \mathbb{R}$ such that $h$ is $L_h$-smooth on $\mathbb{R}^d$. By following Algorithm 3 with step size $\alpha \le \frac{1}{4 L_h (k+2)}$, we have for all $t \ge 1$:*

$$\frac{\alpha}{4} \, \mathbb{E}\big[\|\nabla h(\theta^t)\|_2^2\big] \le \mathbb{E}\big[h(\theta^t) - h(\theta^{t+1})\big] + \frac{L_h \, k \, \mu^2}{16}.$$

---

By averaging Lemma E.1 over $t = 1, \ldots, T$ while keeping a fixed smoothing radius $\mu$.

---

**Theorem E.2.** *Assume the observed objective is of the form $h(x) = g(Ax)$ for some matrix $A \in \mathbb{R}^{k \times d}$ with full row rank $\text{rank}(A) = k \le d$, and some function $g : \mathbb{R}^k \to \mathbb{R}$ such that $h$ is $L_h$-smooth on $\mathbb{R}^d$ and bounded below, i.e., $h^\star := \inf_{\theta \in \mathbb{R}^d} h(\theta) > -\infty$. Define $\Delta h := h(\theta^1) - h^\star$ and let $T \ge 1$. By following Algorithm 3 with step size $\alpha \le \frac{1}{4 L_h (k+2)}$, we have:*

$$\frac{1}{T} \sum_{t=1}^{T} \mathbb{E}\big[\|\nabla h(\theta^t)\|_2^2\big] \le \frac{4 \, \Delta h}{\alpha \, T} + \frac{L_h \, k \, \mu^2}{4 \, \alpha}.$$

*In particular, for the maximal admissible stepsize $\alpha = \frac{1}{4 L_h (k+2)}$, we have:*

$$\frac{1}{T} \sum_{t=1}^{T} \mathbb{E}\big[\|\nabla h(\theta^t)\|_2^2\big] \le \frac{16 \, L_h \, (k + 2) \, \Delta h}{T} + L_h^2 \, k \, (k + 2) \, \mu^2.$$

---

*Remark* E.3. The bound in Theorem E.2 exhibits the usual tradeoff induced by a *fixed* smoothing radius $\mu$: the averaged squared gradient decreases as $\mathcal{O}(k/T)$ up to a non-vanishing bias floor of order $\mathcal{O}(k^2 \mu^2)$. Therefore, for any target accuracy $\varepsilon > 0$, if we choose $\alpha = \frac{1}{4 L_h (k+2)}$ and the smoothing parameter such that $L_h^2 \, k \, (k + 2) \, \mu^2 \le \frac{\varepsilon^2}{2}$ and take

$T \geq \frac{32\, L_h\, (k+2)\, \Delta h}{\varepsilon^2}$, then $\frac{1}{T} \sum_{t=1}^{T} \mathbb{E}[\|\nabla h(\theta^t)\|_2^2] \leq \varepsilon^2$. By Jensen's inequality, $\frac{1}{T} \sum_{t=1}^{T} \mathbb{E}[\|\nabla h(\theta^t)\|_2] \leq \varepsilon$ under the same choice. Hence, as for NCRS, the number of function evaluations required to reach an averaged stationarity level $\varepsilon$ scales as $\mathcal{O}(k/\varepsilon^2)$ in the ridge model, provided that the smoothing radius $\mu$ is chosen sufficiently small.

### E.2. Proofs

**Lemma E.4.** *Let $P \in \mathbb{R}^{d \times d}$ be an orthogonal projector, i.e., $P^\top = P$ and $P^2 = P$, with $\mathrm{rank}(P) = k$. Let $z \sim \mathcal{N}(0, P)$. Then*

$$\begin{cases} \mathbb{E}\|z\|_2^2 = k, \\ \mathbb{E}\|z\|_2^4 = k(k+2), \\ \mathbb{E}\|z\|_2^6 = k(k+2)(k+4). \end{cases}$$

*Proof.* Since $P$ is an orthogonal projector of rank $k$, there exists a matrix $U \in \mathbb{R}^{d \times k}$ with orthonormal columns such that

$$P = UU^\top.$$

Let $\xi \sim \mathcal{N}(0, I_k)$ and define $z := U\xi$. Then $z$ is Gaussian with mean $0$ and covariance $UU^\top = P$, hence $z \sim \mathcal{N}(0, P)$.

Moreover, because $U$ has orthonormal columns,

$$\|z\|_2^2 = \|U\xi\|_2^2 = \xi^\top U^\top U \xi = \|\xi\|_2^2.$$

Therefore $\mathbb{E}\|z\|_2^2 = \mathbb{E}\|\xi\|_2^2 = k$, $\quad \mathbb{E}\|z\|_2^4 = \mathbb{E}\|\xi\|_2^4 = k(k+2)$ and $\quad \mathbb{E}\|z\|_2^6 = \mathbb{E}\|\xi\|_2^6 = k(k+2)(k+4)$. $\qquad \square$

**Lemma E.5.** *Let $P \in \mathbb{R}^{d \times d}$ be an orthogonal projector, i.e., $P^\top = P$ and $P^2 = P$, with $\mathrm{rank}(P) = k$. Let $s \sim \mathcal{N}(0, I_d)$. Then for any $a \in \mathbb{R}^d$,*

$$\mathbb{E}\Big[(a^\top s)^2\, \|Ps\|_2^2\Big] = k\,\|a\|_2^2 + 2\, a^\top P a.$$

*In particular, if $a \in \mathrm{range}(P)$, then*

$$\mathbb{E}\Big[(a^\top s)^2\, \|Ps\|_2^2\Big] = (k+2)\|a\|_2^2.$$

*Proof.* Write $\|Ps\|_2^2 = s^\top P s = \mathrm{tr}(Pss^\top)$, hence

$$(a^\top s)^2 \|Ps\|_2^2 = (a^\top s)^2\, \mathrm{tr}(Pss^\top) = \mathrm{tr}\Big(P\, (a^\top s)^2\, ss^\top\Big).$$

Taking expectation and using linearity of trace,

$$\mathbb{E}[(a^\top s)^2 \|Ps\|_2^2] = \mathrm{tr}\Big(P\, \mathbb{E}[(a^\top s)^2\, ss^\top]\Big).$$

We now compute the matrix $M := \mathbb{E}[(a^\top s)^2\, ss^\top]$ entrywise. For $i, j \in \{1, \ldots, d\}$,

$$M_{ij} = \mathbb{E}[(a^\top s)^2 s_i s_j] = \sum_{p,q=1}^{d} a_p a_q\, \mathbb{E}[s_p s_q s_i s_j].$$

By *Isserlis' theorem* (Wick's formula) for centered Gaussian vectors, for $s \sim \mathcal{N}(0, I_d)$ we have, for all indices $p, q, i, j$,

$$\mathbb{E}[s_p s_q s_i s_j] = \delta_{pq}\delta_{ij} + \delta_{pi}\delta_{qj} + \delta_{pj}\delta_{qi},$$

where $\delta$ denotes the Kronecker delta. Plugging this in gives

$$M_{ij} = \Big(\sum_{p=1}^{d} a_p^2\Big)\delta_{ij} + a_i a_j + a_j a_i = \|a\|_2^2\, \delta_{ij} + 2a_i a_j.$$

Hence $M = \|a\|_2^2 I_d + 2aa^\top$. Therefore

$$\mathbb{E}\big[(a^\top s)^2 \|Ps\|_2^2\big] = \text{tr}\Big(P\big(\|a\|_2^2 I_d + 2aa^\top\big)\Big) = \|a\|_2^2 \,\text{tr}(P) + 2\,\text{tr}(Paa^\top).$$

Finally, $\text{tr}(P) = \text{rank}(P) = k$ and $\text{tr}(Paa^\top) = a^\top Pa$, yielding

$$\mathbb{E}\big[(a^\top s)^2 \|Ps\|_2^2\big] = k\|a\|_2^2 + 2a^\top Pa.$$

If $Pa = a$, then $a^\top Pa = \|a\|_2^2$, giving the last statement. □

---

**Lemma E.6** (Theorem E.1). *Assume that $h : \mathbb{R}^d \to \mathbb{R}$ is $L_h$-smooth. Assume furthermore that $h(\theta) = g(A\theta)$ for some $A \in \mathbb{R}^{k \times d}$ with $\text{rank}(A) = k$. By following Algorithm 3 with step size $\alpha \le \frac{1}{4L_h(k+2)}$, we have:*

$$\frac{\alpha}{4}\, \mathbb{E}\big[\|\nabla h(\theta^t)\|_2^2\big] \ \le\ \mathbb{E}\big[h(\theta^t) - h(\theta^{t+1})\big] + \frac{L_h k \mu^2}{16}.$$

---

*Proof of Lemma E.1.* For all $\theta \in \mathbb{R}^d$, we denote

$$\begin{cases} \widetilde{g}_t := \dfrac{h(\theta^t + \mu s_t) - h(\theta^t)}{\mu}, \\[2mm] g_t := \langle \nabla h(\theta^t),\, s_t\rangle. \end{cases}$$

Let $t \ge 1$ and let $P := A^\top(AA^\top)^{-1}A$ the orthogonal projector onto $\text{range}(A^\top)$. We have:

$$\begin{aligned}
\mathbb{E}\big[h(\theta^{t+1}) \mid \theta^t\big] &= \mathbb{E}\left[h\left(\theta^t - \alpha\,\frac{h(\theta^t + \mu s_t) - h(\theta^t)}{\mu}\, s_t\right) \,\Big|\, \theta^t\right] \\
&= \mathbb{E}\left[g\left(A\theta^t - \alpha\,\frac{h(\theta^t + \mu s_t) - h(\theta^t)}{\mu}\, As_t\right) \,\Big|\, \theta^t\right] \\
&= \mathbb{E}\left[g\left(A\theta^t - \alpha\,\frac{h(\theta^t + \mu s_t) - h(\theta^t)}{\mu}\, APs_t\right) \,\Big|\, \theta^t\right] \\
&= \mathbb{E}\left[h\big(\theta^t - \alpha\,\widetilde{g}_t\, Ps_t\big) \,\Big|\, \theta^t\right] \\
&\le \mathbb{E}\left[h(\theta^t) - \alpha\,\widetilde{g}_t\,\langle\nabla h(\theta^t), Ps_t\rangle + \frac{L_h}{2}\big(\alpha\,\widetilde{g}_t\,\|Ps_t\|_2\big)^2 \,\Big|\, \theta^t\right] \quad \text{(smoothness)} \\
&= \mathbb{E}\left[h(\theta^t) - \alpha\,\widetilde{g}_t\,\langle P^\top\nabla h(\theta^t), s_t\rangle + \frac{L_h}{2}\big(\alpha\,\widetilde{g}_t\,\|Ps_t\|_2\big)^2 \,\Big|\, \theta^t\right] \\
&= \mathbb{E}\left[h(\theta^t) - \alpha\,\widetilde{g}_t\,\langle P\nabla h(\theta^t), s_t\rangle + \frac{L_h}{2}\big(\alpha\,\widetilde{g}_t\,\|Ps_t\|_2\big)^2 \,\Big|\, \theta^t\right] \quad (P = P^\top) \\
&= \mathbb{E}\left[h(\theta^t) - \alpha\,\widetilde{g}_t g_t + \frac{L_h}{2}\big(\alpha\,\widetilde{g}_t\big)^2 \|Ps_t\|_2^2 \,\Big|\, \theta^t\right]. \quad \text{(see Equation (2))}
\end{aligned}$$

Write $\delta_t := g_t - \widetilde{g}_t$. Then

$$-\widetilde{g}_t\, g_t = -(g_t - \delta_t)g_t = -g_t^2 + \delta_t g_t \le -g_t^2 + \frac{1}{2}g_t^2 + \frac{1}{2}\delta_t^2 = -\frac{1}{2}\,g_t^2 + \frac{1}{2}\,\delta_t^2.$$

Using the smoothness of $h$, we also have the following bound:

$$\begin{aligned}
\Big|h(\theta^t + \mu s_t) - h(\theta^t) - \mu\langle\nabla h(\theta^t), s_t\rangle\Big| &= \Big|h(\theta^t + \mu Ps_t) - h(\theta^t) - \mu\langle\nabla h(\theta^t), Ps_t\rangle\Big| \\
&\le \frac{L_h}{2}\,\mu^2\|Ps_t\|^2.
\end{aligned}$$

This implies that $\delta_t^2 \leq \frac{L_h^2 \mu^2}{4} \|Ps_t\|^4$. Thus $-\widetilde{g}_t \, g_t \leq -\frac{g_t^2}{2} + \frac{L_h^2 \mu^2}{8} \|Ps_t\|^4$. We deduce that:

$$\mathbb{E}[h(\theta^{t+1}) \,|\, \theta^t] \leq h(\theta^t) - \frac{\alpha}{2} \mathbb{E}[g_t^2 \,|\, \theta^t] + \frac{\alpha L_h^2 \mu^2}{8} \mathbb{E}\|Ps_t\|^4 + \frac{L_h}{2} \alpha^2 \mathbb{E}[\widetilde{g}_t^2 \|Ps_t\|^2 \,|\, \theta^t].$$

We remark that:

$$\widetilde{g}_t^2 \|Ps_t\|^2 \leq 2g_t^2 \|Ps_t\|^2 + 2\delta_t^2 \|Ps_t\|^2$$

$$\leq 2g_t^2 \|Ps_t\|^2 + \frac{L_h^2 \mu^2}{2} \|Ps_t\|^6.$$

Therefore:

$$\mathbb{E}[h(\theta^{t+1}) \,|\, \theta^t] \leq h(\theta^t) - \frac{\alpha}{2} \mathbb{E}[g_t^2 \,|\, \theta^t] + \frac{\alpha L_h^2 \mu^2}{8} \mathbb{E}\|Ps_t\|^4 + L_h \alpha^2 \mathbb{E}[g_t^2 \|Ps_t\|^2 \,|\, \theta^t] + \frac{L_h^3 \mu^2 \alpha^2}{4} \mathbb{E}\|Ps_t\|^6.$$

Using Theorem E.4 and Theorem E.5, we obtain:

$$\mathbb{E}[h(\theta^{t+1}) \,|\, \theta^t] \leq h(\theta^t) - \frac{\alpha}{2} \mathbb{E}[g_t^2 \,|\, \theta^t] + \frac{\alpha L_h^2 k(k+2)\mu^2}{8} + L_h \alpha^2 (k+2) \|\nabla h(\theta^t)\|_2^2 + \frac{L_h^3 \mu^2 \alpha^2 k(k+2)(k+4)}{4}.$$

Let $e_1 = (1, 0, \ldots, 0) \in \mathbb{R}^d$. We have:

$$\mathbb{E}[g_t^2 \,|\, \theta^t] = \mathbb{E}[\langle s_t, e_1 \rangle^2 \|\nabla h(\theta^t)\|_2^2 \,|\, \theta^t] = \|\nabla h(\theta^t)\|_2^2 \mathbb{E}_{s \sim \mathcal{N}(0,1)} s^2 = \|\nabla h(\theta^t)\|_2^2.$$

It follows that:

$$\left( \frac{\alpha}{2} - L_h (k+2) \alpha^2 \right) \mathbb{E}\|\nabla h(\theta^t)\|_2^2 \leq \mathbb{E}[h(\theta^t) - h(\theta^{t+1})] + \frac{\alpha \, L_h^2 \, k(k+2) \, \mu^2}{8} + \frac{L_h^3 \, \alpha^2 \, k(k+2)(k+4) \, \mu^2}{4}.$$

Since $\alpha \leq \frac{1}{4L_h(k+2)}$, it holds that $\alpha L_h(k+2) \leq \frac{1}{4}$, hence

$$\frac{\alpha}{2} - L_h(k+2)\alpha^2 = \alpha \left( \frac{1}{2} - \alpha L_h(k+2) \right) \geq \frac{\alpha}{4}.$$

This implies that:

$$\frac{\alpha}{4} \mathbb{E}\|\nabla h(\theta^t)\|_2^2 \leq \mathbb{E}[h(\theta^t) - h(\theta^{t+1})] + \frac{\alpha L_h^2 k(k+2)\mu^2}{8} + \frac{L_h^3 \alpha^2 k(k+2)(k+4)\mu^2}{4}.$$

Moreover, we have also:

$$\frac{\alpha L_h^2 k(k+2)\mu^2}{8} \leq \frac{L_h k \mu^2}{32} \quad \text{and} \quad \frac{L_h^3 \alpha^2 k(k+2)(k+4)\mu^2}{4} \leq \frac{L_h k \mu^2}{32},$$

so that:

$$\frac{\alpha}{4} \mathbb{E}\|\nabla h(\theta^t)\|_2^2 \leq \mathbb{E}[h(\theta^t) - h(\theta^{t+1})] + \frac{L_h k \mu^2}{16}.$$

$\square$

## F. From Probabilistic Link Functions to Our Confidence-Score Model

Let $\Delta(x, y) := f(x) - f(y)$. A classical pairwise-comparison model is specified by a *probabilistic link* $\sigma : \mathbb{R} \to [0, 1]$ such that, conditionally on $(x, y)$, the oracle outputs a label $B(x, y) \in \{-1, +1\}$ with:

$$\Pr(B(x, y) = +1 \,|\, x, y) = \sigma(\Delta(x, y)),$$

where $B = +1$ means "prefer $y$ over $x$" and $B = -1$ means the opposite. Typically $\sigma$ is nondecreasing, satisfies $\sigma(0) = \frac{1}{2}$, and is *antisymmetric*: $\sigma(-u) = 1 - \sigma(u)$ for all $u \in \mathbb{R}$.

**A confidence-score oracle satisfying Assumption 1.2.** Given such a link $\sigma$, we define the deterministic confidence score:

$$\tilde{R}(x,y) \;:=\; 2\sigma(\Delta(x,y)) - 1 \;\in [-1,1].$$

Define $\rho : \mathbb{R}_+ \to [0,1]$ by:

$$\forall t \geq 0, \; \rho(t) \;:=\; 2\sigma(t) - 1.$$

Then $\rho(0) = 0$ and $\rho$ is nondecreasing on $\mathbb{R}_+$. Moreover, by antisymmetry, for any $u \in \mathbb{R}$, we have:

$$2\sigma(u) - 1 = \text{sign}(u)\left(2\sigma(|u|) - 1\right) = \text{sign}(u)\,\rho(|u|).$$

Applying this with $u = \Delta(x,y)$ gives, for $\Delta(x,y) \neq 0$,

$$\text{sign}(\Delta(x,y))\,\tilde{R}(x,y) = \text{sign}(\Delta(x,y))\left(2\sigma(\Delta(x,y)) - 1\right) = \rho(|\Delta(x,y)|).$$

Since $\tilde{R}(x,y)$ is deterministic given $(x,y)$, we have:

$$\mathbb{E}\Big[\text{sign}(\Delta(x,y))\,\tilde{R}(x,y) \mid x,y\Big] = \rho(|\Delta(x,y)|).$$

Furthermore,

$$\mathbb{E}\Big[\tilde{R}(x,y)^2 \mid x,y\Big] = \tilde{R}(x,y)^2 = \rho(|\Delta(x,y)|)^2 \leq \rho(|\Delta(x,y)|),$$

because $\rho(\cdot) \in [0,1]$. Therefore the first two conditions of Assumption 1.2 are satisfied with $C = 1$.

**Local linear growth of $\rho$ near $0$.** Assumption 1.2 further requires the existence of constants $c > 0$ and $r > 0$ such that $\rho(t) \geq ct$ for all $t \in [0,r]$. This property follows from a mild regularity condition on the link $\sigma$.

---

**Lemma F.1.** *Assume that $\sigma$ is differentiable at $0$ and $\sigma'(0) > 0$. Then there exist constants $c > 0$ and $r > 0$ such that for all $t \in [0,r]$, we have*

$$\rho(t) = 2\sigma(t) - 1 \;\geq\; ct.$$

*In particular, one may take $c = \sigma'(0)$ and $r > 0$ small enough.*

---

*Proof.* Since $\sigma$ is differentiable at $0$ with $\sigma(0) = 1/2$, we have the first-order expansion $\sigma(t) = \frac{1}{2} + \sigma'(0)\,t + o(t)$ as $t \downarrow 0$. Hence:

$$\rho(t) = 2\sigma(t) - 1 = 2\sigma'(0)\,t + o(t).$$

Choose $r > 0$ such that $|o(t)| \leq \sigma'(0)\,t$ for all $t \in [0,r]$. Then for $t \in [0,r]$, we have:

$$\rho(t) \;\geq\; (2\sigma'(0) - \sigma'(0))\,t \;=\; \sigma'(0)\,t.$$

$\square$

**Classical examples.** We list standard probabilistic links $\sigma : \mathbb{R} \to [0,1]$ used in pairwise comparison models and verify that they satisfy the condition of Lemma F.1.

- **Logistic / Bradley–Terry.** For $\tau > 0$, define:

$$\sigma(u) \;:=\; \frac{1}{1 + e^{-u/\tau}}.$$

  Then $\sigma(0) = \frac{1}{2}$, $\sigma$ is nondecreasing, and $\sigma(-u) = 1 - \sigma(u)$. Moreover,

$$\sigma'(u) = \frac{1}{\tau}\,\sigma(u)\big(1 - \sigma(u)\big) \quad \Rightarrow \quad \sigma'(0) = \frac{1}{4\tau} > 0.$$

- **Probit / Thurstone.** For $\sigma_0 > 0$, define:

$$\sigma(u) := \Phi\left(\frac{u}{\sigma_0}\right),$$

where $\Phi$ is the standard normal CDF. Then $\sigma(0) = \frac{1}{2}$, $\sigma$ is nondecreasing, and $\sigma(-u) = 1 - \sigma(u)$ by symmetry of the normal distribution. Since $\Phi'(z) = \varphi(z)$ where $\varphi(z) = \frac{1}{\sqrt{2\pi}} e^{-z^2/2}$ is the standard normal pdf, we have

$$\sigma'(u) = \frac{1}{\sigma_0} \varphi\left(\frac{u}{\sigma_0}\right) \quad \Rightarrow \quad \sigma'(0) = \frac{1}{\sigma_0 \sqrt{2\pi}} > 0.$$

- **Arctan.** For $\tau > 0$, define:

$$\sigma(u) := \frac{1}{2} + \frac{1}{\pi} \arctan\left(\frac{u}{\tau}\right).$$

Then $\sigma(0) = \frac{1}{2}$, $\sigma$ is nondecreasing, and $\sigma(-u) = 1 - \sigma(u)$. Moreover,

$$\sigma'(u) = \frac{1}{\pi\tau} \cdot \frac{1}{1 + (u/\tau)^2} \quad \Rightarrow \quad \sigma'(0) = \frac{1}{\pi\tau} > 0,$$

**Resulting $\rho$ for these links.** With $\rho(t) = 2\sigma(t) - 1$, the above examples yield, respectively, $\rho(t) = \tanh\left(\frac{t}{2\tau}\right)$, $\rho(t) = 2\Phi(t/\sigma_0) - 1$, and $\rho(t) = \frac{2}{\pi} \arctan(t/\tau)$, all of which satisfy $\rho(t) \geq ct$ for $t$ in a neighborhood of 0 by Lemma F.1.

## G. Controlled synthetic experiment: isolating the intrinsic-dimension scaling

We include a controlled synthetic experiment to isolate the dependence on the intrinsic dimension $k$ predicted by the theory. Unlike the language-model and reinforcement-learning experiments, this experiment exactly matches the ridge setting analyzed in the paper and keeps the ambient dimension fixed.

We fix the ambient dimension to $d = 500$ and vary the intrinsic dimension over

$$k \in \{5, 10, 20, 40, 80\}.$$

For each value of $k$, we sample a matrix $A_k \in \mathbb{R}^{k \times d}$ with orthonormal rows and consider the ridge objective

$$f_k(x) = g(A_k x) \quad \text{with} \quad g(z) = \sum_{i=1}^{k} \left(1 - \cos(z_i)\right).$$

This objective is 1-smooth. Indeed,

$$\nabla^2 f_k(x) = A_k^\top \operatorname{diag}\left(\cos(A_k x)\right) A_k,$$

and since $A_k A_k^\top = I_k$, its operator norm is at most 1.

To make the optimization instances comparable across different values of $k$, we initialize

$$x_0 = A_k^\top z_0, \qquad z_0 = \left(\arcsin(1/\sqrt{k}), \ldots, \arcsin(1/\sqrt{k})\right).$$

Then $A_k x_0 = z_0$, and therefore

$$\|\nabla f_k(x_0)\|_2 = \|\nabla g(z_0)\|_2 = \left\|\left(\frac{1}{\sqrt{k}}, \ldots, \frac{1}{\sqrt{k}}\right)\right\|_2 = 1.$$

We run NCRS with target stationarity accuracy $\epsilon = 0.1$ under the two oracle models considered in the paper. For the uniform-margin oracle, each iteration uses one comparison. For the confidence-oracle model, we use the fixed-vote schedule

$$N = \left\lceil \frac{k}{\epsilon^2} \right\rceil$$

comparisons per iteration, as suggested by the theory. In both cases, we use a $k$-normalized stepsize of the form

$$\alpha_k = \frac{c}{k},$$

with the same constant $c = 0.8$ for all values of $k$.

The results are reported in Table 5. Under the uniform-margin oracle, the number of comparisons grows approximately linearly with $k$, consistent with the $O(k/(p^2\epsilon^2))$ complexity. Under the confidence oracle, the number of iterations also grows approximately linearly with $k$, while the total number of comparisons grows approximately quadratically with $k$, consistent with the $O(k^2/\epsilon^4)$ complexity. This experiment is intended as a controlled validation of the theoretical intrinsic-dimension scaling, rather than as a practical efficiency comparison for fixed-vote confidence aggregation.

*Table 5.* Controlled synthetic ridge experiment with fixed ambient dimension $d = 500$ and varying intrinsic dimension $k$. The uniform-margin oracle uses one comparison per iteration. The confidence oracle uses $N = \lceil k/\epsilon^2 \rceil$ votes per iteration with $\epsilon = 0.1$.

| $k$ | Uniform margin comparisons | Confidence oracle iterations | Confidence oracle comparisons |
|---|---|---|---|
| 5 | 898 | 340 | 170000 |
| 10 | 1780 | 695 | 695000 |
| 20 | 3620 | 1346 | 2691000 |
| 40 | 7414 | 2687 | 10748000 |
| 80 | 14940 | 5414 | 43312000 |

