# OpenReview forum: "Noisy Pairwise-Comparison Random Search for Smooth Nonconvex Optimization"
_ICML.cc/2026/Conference — ICML 2026 regular_

### Official Review · Reviewer_u1LA · 2026-03-04

**Soundness:** 3
**Presentation:** 1
**Significance:** 2
**Originality:** 2
**Overall Recommendation:** 3
**Confidence:** 3

**Summary:**

The authors propose a simple zeroth-order algorithm, noisy-comparison random search (NCRS), for smooth nonconvex optimization problems under comparison oracles. The authors also provides complexity guarantees for NCRS that depends on the intrinsic dimension under the uniform-margin noise model and the confidence oracle model. The empirical results show that NCRS performs competitively against several baselines.

**Compliance With Llm Reviewing Policy:**

Affirmed.

**Final Justification:**

The theorems rely on strong assumptions, and in more general settings, the constants in those assumptions are often unknown. The setting considered in the paper also seems somewhat narrow. In addition, the paper appears to have been written in haste and would require substantial revision. I am not confident that revision would necessarily make it significantly more readable. For these reasons, I would keep my original score unchanged.

**Key Questions For Authors:**

1. The authors wrote in abstract that NCRS uses random line search. Where is the random line search in the algorithm? I cannot find it in the paper.

2. The authors assume $f(x) = g(Ax)$. This assumption is similar to the thought of random projection and random subspace algorithms. What is the difference between the assumption and random subspace algorithm work?

3. If the objective function $f$ does not satisfy $L$-smoothness, can we estimate $L$ under comparison oracles?

**Limitations:**

No. The authors should discuss the limitations and potential negative societal impact.

**Strengths And Weaknesses:**

***Strengths***

- The authors propose a simple zeroth-order algorithm, NCRS, that operates using only a noisy comparison oracle and achieves improved complexity guarantees by exploiting low intrinsic dimensionality.

- Beyond the uniform-margin ranking oracle model (Assumption 1.1), the paper also provides a convergence analysis under the more realistic, gap-dependent confidence oracle model (Assumption 1.2).

- The empirical results are generally positive: while some baselines outperform NCRS on certain tasks, NCRS performs competitively overall and supports the paper's claim.

***Weaknesses***

1. The manuscript is not self-contained and lacks basic problem setup and notation, which gives the impression of being written in haste. In particular, the optimization problem itself (e.g., $\min_{x\in\mathbb{R}^d} f(x)$) and the definition/assumptions on $f$ are not stated clearly early on (e.g., $f$ appears before being defined around p. 1, l. 11). Several abbreviations are used without being expanded at first occurrence (e.g., CBO, ZO, SGD). Moreover, $L$-smoothness and the descent lemma should be explicitly stated at least in Appendix.

2. The theoretical development is restricted to objective functions of the ridge/active-subspace form $f(x)=g(Ax)$. This structural assumption can be strong in practice, and scenarios where the active subspace is unknown or varies with $x$ are not clearly addressed.

3. The step-size choices required to obtain the stated rates depend explicitly on the intrinsic dimension $k$, which is typically unknown in applications. The authors does not discuss stepsizes when $k$ is unknown.

---

> ### Author Rebuttal · Authors · 2026-03-30
>
> We thank the reviewer for the thoughtful feedback and helpful questions.
>
> **Weakness 1.** We thank the reviewer for this helpful comment. We agree that the presentation can be made more self-contained and that a few quantities and abbreviations are introduced before being defined as clearly as they should be. In the revision, we will define the key quantities more clearly before first use, expand all abbreviations (e.g., CBO, ZO, SGD) at first occurrence, and state the optimization problem more explicitly at the beginning of the paper. We will also state the $L$-smoothness assumption explicitly in the main text, and include the standard quadratic upper bound given by the smoothness inequality in the appendix. We appreciate this point and will use the revision to improve the clarity of the presentation.
>
> **Weakness 2.** We note that the algorithm itself does not assume that the active subspace or the matrix $A$ is known: the algorithm is run directly in the ambient space, and $A$ appears only in the analysis as a way to formalize low-dimensional structure. For a detailed discussion of the restrictiveness of the ridge model, including our motivation for the fixed-subspace abstraction and a partial robustness extension beyond exact ridge, we refer the reviewer to Weakness (b) in our reply to Reviewer 14aE.
>
> **Weakness 3.**  We agree that the dependence of the theorem-level step size on $k$ deserves to be clarified, especially since $k$ may be unknown in practice, and we will make this explicit in the revision. Importantly, NCRS itself does not require knowledge of the active subspace or of $k$; this dependence appears only in the proof-level tuning, much like the smoothness constant $L$ in the standard gradient-descent condition $\alpha \le 1/L$ for $L$-smooth objectives. One also cannot simply take $\alpha$ arbitrarily small, since the bound contains the usual tradeoff between a term of order $1/\alpha$ and a term of order $\alpha$. In the deterministic setting, this can be handled by an adaptive multiscale variant of NCRS that tests a dyadic grid of radii. Such a grid ensures that one tested radius lies within a factor of $2$ of the proof-optimal choice, and the same smoothness argument as in Theorem 2.2 recovers the same $k/\varepsilon^2$ dependence up to constants, with only an additional logarithmic factor in the number of tested scales. We will mention this dyadic-grid variant explicitly in the paper. A fully rigorous stochastic analysis of this adaptive tuning remains an interesting direction beyond the scope of the current paper.
>
> **Question 1.** We thank the reviewer for pointing this out. The phrase "random line search" in the abstract is imprecise. NCRS does not perform a classical line search. What we intended to convey is that, at each iteration, the method samples a random search direction and probes a single candidate point along that direction. In the revision, we will replace "random line search" by the more accurate description "random comparison search".
>
> **Question 2.**  The two are different. In our paper, $f(x)=g(Ax)$ is a structural assumption on the objective. By contrast, random projection / random subspace methods are algorithmic choices: they explicitly optimize over a chosen low-dimensional subspace. Importantly, NCRS does not assume that $A$ is known and does not project the iterate onto a subspace. The method is run directly in the ambient space.  In this sense, our work is closer to an intrinsic-dimension analysis of a full-space method than to a random-subspace algorithm.
>
> **Question 3.** Our current framework is specific to the $L$-smooth setting, and we do not claim a nonsmooth extension in this paper. The main difficulty is that, in a pure comparison-only model, the oracle is invariant under a strictly increasing transformation $\phi$ of the objective: $f$ and $\phi \circ f$ induce exactly the same rankings, but their smoothed gradients are generally different. Thus, a stationarity notion based on the gradient of a smoothed objective is not the natural target in the non-smooth setting when one has access to only a pure comparison oracle.  For this reason, extending the theory beyond $L$-smoothness is nontrivial and would require additional structure beyond pure comparisons. In richer feedback models, for example when confidence information or a known link function provides calibrated information about local value gaps, smoothing-based nonsmooth targets become much more natural.
>
> **Limitations and Societal impact.** We will add a limitation paragraph noting that the intrinsic-dimension bounds rely on a fixed reference low-dimensional structure, on $L$-smoothness, and on theory-driven choices of step sizes and algorithmic parameters.
> Regarding impact, this work is primarily theoretical, and we do not see a direct negative societal impact specific to the present results.

---

> > ### Author Rebuttal · Reviewer_u1LA · 2026-04-01
> >
> > Thanks for your rebuttal. In Question 3, by saying that $f$ does not satisfy $L$-smoothness, we do not mean that $f$ is nonsmooth. Even when $f$ is continuously differentiable, there are many cases in which its gradient is not Lipschitz continuous. It is precisely this situation, where the gradient is not Lipschitz continuous, that we have in mind in this question. In such a case, how should $L$ be estimated under comparison oracles?

---

> > > ### Author Response · Authors · 2026-04-01
> > >
> > > We thank the reviewer for the quick follow-up and for the clarification. We agree that “not $L$-smooth” does not mean nonsmooth: $f$ may be $C^1$ while $\nabla f$ is not Lipschitz. In that regime, however, the usual global smoothness constant $L$ is no longer defined in the standard sense, so we are not fully sure what object the reviewer has in mind by “estimating $L$” under comparison oracles in this setting. We therefore understand the reviewer’s question as asking what weaker regularity condition, and what corresponding conclusion, could replace the standard $L$-smooth analysis.
> > >
> > > We emphasize that this regime is not part of the results claimed in the present paper, and we mention it here only to address the reviewer’s clarification. If $f$ is assumed to be only $C^1$ and not necessarily $L$-smooth, while exact comparisons are available and the initial sublevel set is bounded, one can aim for a meaningful stationarity conclusion of the form $\liminf_{t\to\infty}||\nabla f(\theta^t)||_2=0$ almost surely.  For simplicity in the proof sketch, we assume that the search directions are normalized and hence uniformly distributed on the unit sphere. Let $K$ denote the initial sublevel set, and for $\alpha>0$ define $\mu(\alpha)$ as the supremum of $||\nabla f(\theta+t\alpha s)-\nabla f(\theta)||_2$ over all $\theta \in K$, $s\in \mathbb{S}^{d-1}$, and $t\in[0,1]$. By the Heine-Cantor theorem, we can show that
> > > $\mu(\alpha)\to 0$ as $\alpha \downarrow 0$. By the fundamental theorem of calculus, we can show that for all $\theta,s$, we have: $ f(\theta+\alpha s)-f(\theta) \le \alpha \langle \nabla f(\theta),s \rangle+\alpha \mu(\alpha).$ Let  $A(\theta)$ be the set of vectors $s$ on the unit sphere such that  $\langle \nabla f(\theta),s \rangle \le - \frac{1}{2\sqrt{d}} ||\nabla f(\theta)||_2$. For fixed $\theta$, by sampling $s$ uniformly over the unit sphere, we can show that the probability that  $s \in A(\theta)$ is at least $\frac{1}{4}.$ We can then prove that for all $\epsilon>0,$ there exists $r(\epsilon)>0$ such that, for all $\theta \in K$ all $s \in A(\theta)$ and all $\alpha \in (0,r(\epsilon)),  || \nabla f(\theta)||_2 \ge \epsilon \Longrightarrow \quad f(\theta + \alpha s) \le f(\theta) - \frac{1}{4\sqrt{d}}  \alpha  ||\nabla f(\theta)||_2.$ Fix $\epsilon >0$. We can deduce that for all $t$ such that $\alpha_t \le r(\epsilon)$, we have: $$\mathbb{E}[f(\theta^{t+1}) | \theta^t] \le  f(\theta^t)-\frac{\alpha_t}{16 \sqrt{d}} || \nabla f(\theta^t)||_2 1({ || \nabla f(\theta^t)||_2}\ge \epsilon).$$ With this, we can prove that if $\sum \alpha_t=\infty$ and $\alpha_t \downarrow 0$, then, almost surely, we have:
> > >
> > > $$\liminf_{t\to\infty}||\nabla f(\theta^t)||_2=0.$$
> > >
> > > We stress, however, that this discussion is only meant to address the reviewer’s question: the present paper focuses on the $L$-smooth setting, and the $C^1$ but non-$L$-smooth regime lies outside its formal scope.
> > >
> > > **If instead the reviewer had in mind the case where $f$ is still $L$-smooth but the constant $L$ is unknown, then this corresponds to the issue already discussed in our response to Weakness 3: in that setting, a multiscale procedure, for example a dyadic grid of radii/step sizes, can be used and recovers the same dependence up to constants and additional logarithmic factors without requiring prior knowledge of $L$.**

---

### Official Review · Reviewer_dDpa · 2026-03-07

**Soundness:** 3
**Presentation:** 2
**Significance:** 3
**Originality:** 3
**Overall Recommendation:** 3
**Confidence:** 3

**Summary:**

This paper studies smooth nonconvex optimization under noisy pairwise-comparison (ordinal) feedback. It proposes Noisy-Comparison Random Search (NCRS), a simple random-direction accept/reject method, and shows that under an active-subspace/intrinsic-dimension model the oracle complexity depends on the intrinsic dimension $k$ rather than the ambient dimension $d$. Under a uniform-margin comparison oracle, NCRS reaches $\epsilon$-stationarity with complexity $O(k/(p^2\epsilon^2))$; under a gap-dependent confidence oracle (degrading near ties), a confidence-weighted/voting variant achieves $\epsilon$-stationarity with complexity $O(k^2/\epsilon^4)$. The paper also reports experiments on masked language model fine-tuning and simulated preference-based RL demonstrating empirical benefits of the proposed approach.

**Compliance With Llm Reviewing Policy:**

Affirmed.

**Key Questions For Authors:**

- Varying active subspace: How sensitive are the guarantees to the assumption that the active subspace is fixed? Can you extend the analysis (even partially) to slowly varying subspaces?
- Tightness of the confidence-oracle rate: Is the $O(k^2/\epsilon^4)$ complexity under the confidence oracle tight? Are there lower bounds or examples suggesting the extra $k/\epsilon^2$ factor is unavoidable?
- Practical parameter selection: In practice, how are step sizes and (for voting) the number of repeated comparisons chosen without knowledge of problem constants? Do you have an adaptive rule that works robustly?

**Limitations:**

Yes

**Strengths And Weaknesses:**

Strengths.
- Soundness: The paper provides clear, explicit intrinsic-dimension guarantees for noisy comparison-based nonconvex optimization, including a uniform-margin model and a confidence-degrading model, with stated $\epsilon$-stationarity rates.
- Originality/Significance: Replacing ambient dimension $d$ with intrinsic dimension $k$ in oracle complexity is potentially impactful for large-scale preference or black-box optimization settings where only comparisons are available.
- Presentation: The proposed method is extremely simple (random direction + accept/reject), making the algorithm and the high-level intuition easy to understand and implement.

Weaknesses.
- Soundness/Scope: The fixed active-subspace assumption is strong; it is unclear how robust the results are when the effective subspace varies along the trajectory.
- Significance (empirics): The experiments do not tightly isolate the predicted $k$-scaling (e.g., controlled studies varying $k$ with fixed $d$), making it difficult to attribute empirical gains to the main theoretical mechanism.
- Originality/Clarity: The algorithmic core is close to classical random/direct search, so novelty rests mainly on assumptions and analysis; in addition, it is unclear whether the $O(k^2/\epsilon^4)$ confidence-oracle rate is tight.
- Writing: what is p in line 23 of abstract? What is CBO in line 47?

---

> ### Author Rebuttal · Authors · 2026-03-30
>
> We thank the reviewer for the careful reading and constructive feedback.
>
> **Weakness 1.**  Please see our response to Weakness (b) in reviewer 14aE’s review, including the partial robustness extension beyond exact ridge structure, which covers controlled non-ridge perturbations.
>
> **Weakness 2.**
> We agree that a controlled study varying $k$ while fixing $d$ isolates the theoretical mechanism more cleanly, and we therefore ran an additional synthetic experiment. We fixed the ambient dimension at $d=500$ and varied the intrinsic dimension over $k\in\{5,10,20,40,80\}$. For each $k$, we sampled a matrix $A_k\in\mathbb{R}^{k\times d}$ with orthonormal rows and considered the exact ridge objective
> $
> f_k(x)=g(A_kx)$, where $g(z)=\sum_{i=1}^k (1-\cos z_i).$
> This objective is $1$-smooth, since
> $
> \nabla^2 f_k(x)=A_k^\top \operatorname{diag}(\cos(A_kx))A_k
> $
> has operator norm at most $1$. We initialized
> $
> x_0=A_k^\top z_0$ where $
> z_0=\bigl(\arcsin(1/\sqrt{k}),\ldots,\arcsin(1/\sqrt{k})\bigr),$
> so that $||\nabla f_k(x_0)||_2=1$ for every $k$. We then ran NCRS under the two oracle models from the paper with target accuracy $\epsilon=0.1$, using the $k$-normalized step-size scaling $\alpha_k=c/k$ with the same constant across all $k$; in the experiment we used $c=0.8$.
> For the uniform-margin  model, each iteration uses exactly one comparison, so the total number of comparisons equals the number of iterations. Empirically, this quantity grows approximately linearly with $k$. For the fixed-vote confidence-oracle  model with $N=\lceil k/\varepsilon^2\rceil$ votes per iteration, the median number of iterations remains approximately linear in $k$ (slope close to $1$), while the median total number of comparisons is approximately quadratic in $k$. Thus, this controlled fixed-$d$ experiment directly isolates the intrinsic-dimension effect and matches the $O(k/(p^2\epsilon^2))$ and $O(k^2/\epsilon^4)$ scaling predicted by the theory.
>
> | $k$ | Uniform margin (comparaisons) | Confidence oracle (iterations) | Confidence oracle (comparaisons) |
> |---|---:|---:|---:|
> | 5  | 898   | 340  | 170000   |
> | 10 | 1780  | 695  | 695000   |
> | 20 | 3620  | 1346 | 2691000  |
> | 40 | 7414  | 2687 | 10748000 |
> | 80 | 14940 | 5414 | 43312000 |
>
> **Weakness 3.** **Novelty:**  We agree that the algorithmic core is intentionally simple. The main contribution of the paper is theoretical: we introduce and analyze noisy comparison-oracle models for smooth nonconvex zeroth-order optimization, and we show how qualitatively different noise mechanisms lead to different complexity scalings. Thus, the novelty is not in the search step by itself, but in the noise modeling, the convergence analysis, and the resulting understanding of how different forms of pairwise-comparison noise affect the complexity of smooth nonconvex zeroth-order optimization.
> **Tightness of the bound:** We first explain the extra factor: The extra $k/\varepsilon^2$ factor comes from the reliability of each comparison near stationarity, not from the search itself. In the uniform-margin model, each comparison is essentially a $\{\pm1\}$-valued noisy sign with mean of order $p$ and variance of order $1$, so making one accept/reject decision reliable requires the usual $O(1/p^2)$ repetitions. In the confidence-oracle model, however, Assumption 1.2 gives both aligned mean and second moment of order $\rho(|\Delta_t|)$. Thus, when the local gap $|\Delta_t|$ is small, both the signal and the variance shrink at the same rate, and a Bernstein-type bound yields an exponent of order $N\rho(|\Delta_t|)$ rather than $N\rho(|\Delta_t|)^2$. Hence one needs only $N\asymp 1/\rho(|\Delta_t|)$ repeated comparisons to make a single decision reliable.   Near an $\varepsilon$-stationary point, balancing first- and second-order terms suggests $|\Delta_t|\asymp \varepsilon^2/k$; since $\rho$ is locally linear near zero, this gives $N\asymp k/\varepsilon^2$. We also note that we now have a rigorous lower-bound proof for the confidence-oracle model showing that the $k^2/\varepsilon^4$ dependence is optimal within this model. We will add this result formally in the revision and are happy to provide details in the discussion if helpful.
>
> **Weakness 4.** Please see our response to Weakness (1) in reviewer u1LA’s review.
> **Practical voting.** A natural practical rule is to repeat a comparison until the empirical average confidence is sufficiently far from zero to trust its sign. This uses few repetitions for large gaps and more near ties. Since Assumption 1.2 makes both the mean and variance scale like $\rho(|\Delta_t|)$, this suggests an adaptive vote count of order $\log(1/\delta_t) / \rho(|\Delta_t|)$, matching our fixed schedule up to logarithmic factors. We do not formalize this variant here, but we will mention it as a natural practical extension. **Practical step size.** Please see our response to Weakness (3) in reviewer u1LA’s review.

---

> > ### Author Rebuttal · Reviewer_dDpa · 2026-04-02
> >
> > Thank you for the response.

---

> > > ### Author Response · Authors · 2026-04-02
> > >
> > > We thank the reviewer for the time and care they devoted to evaluating the paper, as well as for the helpful remarks in their review, which helped us clarify several important points and strengthen the paper. We also thank the reviewer for the follow-up and for indicating that our rebuttal fully resolved the concerns raised in their review. We are glad that the clarifications were helpful and hope this will be reflected in the final evaluation.

---

### Official Review · Reviewer_fS7A · 2026-03-13

**Soundness:** 3
**Presentation:** 3
**Significance:** 2
**Originality:** 2
**Overall Recommendation:** 4
**Confidence:** 3

**Summary:**

This paper studies smooth nonconvex minimization in high-dimensional settings using only noisy pairwise comparison feedback. The authors propose Noisy-Comparison Random Search (NCRS), a direct-search algorithm that performs random directional exploration and adapts to the intrinsic dimension $k \le d$ via random line search. The authors show that NCRS can find an $\epsilon$-stationary point with complexity $O(k/(p^2\epsilon^2))$, replacing the typical dependence on the ambient dimension $d$ with the intrinsic dimension $k$. The paper also proposes a variant that uses majority voting under a gap-dependent confidence-based comparison oracle and establishes a complexity of $O(k^2/\epsilon^4)$.

**Compliance With Llm Reviewing Policy:**

Affirmed.

**Final Justification:**

The authors addressed my main concerns. I maintain my positive score.

**Key Questions For Authors:**

In the gap-dependent confidence model, the sample complexity is of the order of $\epsilon^{-4}$, which is not ideal compared to others in the table. Do you believe this bound is tight for this specific noise model, or is there room for variance reduction techniques to improve this rate?

**Strengths And Weaknesses:**

Strengths:

1. The paper studies optimization with pairwise comparison feedback, which is relevant to settings such as preference learning and reinforcement learning from human feedback.

2. The analysis highlights how comparison-based optimization methods may depend on the intrinsic dimension of the objective rather than the ambient dimension, which is potentially important in high-dimensional machine learning problems.

3. The paper provides convergence guarantees for the proposed algorithm under two oracle models, including a more realistic gap-dependent confidence model.

Weaknesses

1. The proposed NCRS algorithm is essentially a very simple random search method with an accept–reject rule based on pairwise comparisons. Similar ideas appear frequently in zero-order optimization. The main contribution seems to be the analysis under an intrinsic-dimension model rather than a fundamentally new algorithmic idea.

2.   The empirical validation is relatively limited. Specifically, the evaluation lacks benchmarking against stronger, well-established derivative-free optimization methods, such as those in [1] and [2].

References:

[1]. Malladi, Sadhika, et al. "Fine-tuning language models with just forward passes." Advances in Neural Information Processing Systems 36 (2023)

[2]. Zhang, Yihua, et al. "Revisiting zeroth-order optimization for memory-efficient llm fine-tuning: A benchmark." arXiv preprint arXiv:2402.11592 (2024).

---

> ### Author Rebuttal · Authors · 2026-03-30
>
> We thank the reviewer for the helpful feedback and questions.
>
> **Weakness 1.** The main contribution of the paper is theoretical: we introduce and analyze noisy comparison-oracle models for smooth nonconvex zeroth-order optimization, and show how qualitatively different noise mechanisms lead to different complexity scalings. Thus, the novelty is not in the search step alone, but in the noise modeling, the convergence analysis, and the resulting understanding of how different forms of pairwise-comparison noise affect optimization complexity.
>
> **Weakness 2.** Following the reviewer’s suggestion, we also added a MeZO baseline. This comparison uses 30 total runs (6 fine-tuning settings, each with 5 random seeds), with a 2D hyperparameter sweep over the learning rate $\alpha$ and perturbation parameter $\mu$, following the same protocol as for RSGF. The results, reported as mean iterations $\pm$ 95\% confidence interval, are summarized below.
>
> | $k$ | SGD | NCRS | MeZO | RSGF |
> |---|---:|---:|---:|---:|
> | 1609 | $10761 \pm 1309$ | $51058 \pm 1102$ | $42127 \pm 1376$ | $48078 \pm 1730$ |
> | 1200 | $9511 \pm 1296$  | $35598 \pm 1893$ | $33456 \pm 878$  | $40962 \pm 1594$ |
> | 1037 | $10471 \pm 937$  | $26363 \pm 1424$ | $29860 \pm 743$  | $36745 \pm 1074$ |
> | 896  | $10011 \pm 1028$ | $24863 \pm 786$  | $28257 \pm 633$  | $33077 \pm 799$  |
> | 774  | $9061 \pm 745$   | $21478 \pm 1521$ | $24868 \pm 1134$ | $29754 \pm 1424$ |
> | 207  | $7482 \pm 723$   | $15156 \pm 985$  | $19416 \pm 812$  | $25628 \pm 1115$ |
>
> Overall, these results indicate that NCRS remains competitive with MeZO while consistently outperforming RSGF
>
> **Question 1 (bound not ideal compared to others in Table 1).** The $O(k^2/\varepsilon^4)$ rate in the gap-dependent confidence model should not be viewed as directly worse than the other rates in Table 1, because these entries correspond to different oracle models. The relevant like-for-like comparison is the nonconvex comparison-oracle setting. In particular, the deterministic $O(d/\varepsilon^2)$ rate is already recovered in our framework as the noiseless special case of the uniform-margin model with $p=1/2$. By contrast, the confidence-oracle model is strictly harder, since comparison quality deteriorates near ties, which is precisely what produces the additional $k/\varepsilon^2$ factor. The other $O(d/\varepsilon^2)$ rate in the table, from Wang et al., is also not directly comparable: it assumes access to stochastic observations of $f_\xi$, rather than a direct comparison oracle, and its convergence regime requires a **very strong** condition, namely that both the variance of the stochastic function values and the variance of the stochastic gradient samples are already of order $\widetilde O(\varepsilon)$. Thus, Table 1 is intended to compare qualitatively different feedback models, not to suggest that all rates are directly comparable.
>
> **Question 2 (tightness of the bound).** We first explain the exctra factor. The extra $k/\varepsilon^2$ term comes from the reliability of each comparison near stationarity, not from the search itself. In the uniform-margin model, each comparison is essentially a $\{\pm 1\}$-valued noisy sign with mean of order $p$ and variance of order $1$, so making one accept/reject decision reliable requires the usual $O(1/p^2)$ repetitions. In the confidence-oracle model, however, Assumption 1.2 gives both aligned mean and second moment of order $\rho(|\Delta_t|)$. Thus, when the local gap $|\Delta_t|$ is small, both the signal and the variance shrink at the same rate, and a Bernstein-type bound yields an exponent of order $N\rho(|\Delta_t|)$ rather than $N\rho(|\Delta_t|)^2$. Hence one needs only $N\asymp 1/\rho(|\Delta_t|)$ repeated comparisons to make a single decision reliable. Near an $\varepsilon$-stationary point, balancing first- and second-order terms suggests $|\Delta_t|\asymp \varepsilon^2/k$; since $\rho$ is locally linear near zero, this gives $N\asymp k/\varepsilon^2$. We also note that we now have a rigorous lower-bound proof for the confidence-oracle model showing that the $k^2/\varepsilon^4$ dependence is optimal within this model. We will add this result formally in the revision and are happy to provide details in the discussion if helpful.

---

> > ### Author Rebuttal · Reviewer_fS7A · 2026-04-02
> >
> > Thanks for your response. I maintain my positive score.

---

### Official Review · Reviewer_14aE · 2026-03-21

**Soundness:** 3
**Presentation:** 3
**Significance:** 3
**Originality:** 3
**Overall Recommendation:** 4
**Confidence:** 4

**Summary:**

This paper proposes an innovative zero-order optimization algorithm, NCRS, based on a random oracle that returns the sign of the function difference, and establishes a non-convex smooth convergence theoretical guarantee for the algorithm. The algorithm's complexity replaces the dependency on dimension with a weaker intrinsic dimensiona dependence.

**Compliance With Llm Reviewing Policy:**

Affirmed.

**Final Justification:**

I thank the authors for their response. I decided to maintain my overall score.

**Key Questions For Authors:**

I'm not sure if I understand the author's research background. From the perspective of general optimization problems, could the author have more specifically formalized their research into a mathematical framework, incorporating some low-dimensional structural settings?

How general is the linearity of embedding high-dimensional vectors into low-dimensional subspaces, as considered in the article, in real-world optimization scenarios?

**Limitations:**

The author did not discuss some limitations of the work; "weakness" reveals some potential limitations.

**Strengths And Weaknesses:**

Strengths:

(a) The authors' proposed NCRS method, as a zero-order method, when considering low-dimensional structures with monotone observation, exhibits convergence complexity dependent on the dimension k of the embedding space rather than the dimension d of the decision variables. This is advantageous when $k$ is much smaller than $d$.

(b) The authors analyze a confidence-weighted vote variant of NCRS, which has upper complexity that captures the near-stationary regime where comparisons become less informative as the difference in function values shrinks.

(c) The authors empirically validate NCRS on masked language model fine-tuning and preference-based RL benchmarks. This strengthens the research's impact on the machine learning community.

Weakness:

(a) While the stochastic sign-comparison oracle proposed by the authors introduces randomness into the algorithm, it doesn't seem to consider that in large-scale machine learning, we typically only evaluate the loss of a single or mini-batch of samples. When considering evaluating random function values, the critical assumption 1.1, requiring prediction accuracy to exceed 50%, may not be satisfied.

(b) The authors assume that the low-dimensional embedding mapping is linear; I'm unsure if this assumption is too stringent for general problems.

(c) Although the authors haven't included a conclusion section in the main text, I think it's worthwhile to discuss the limitations of this work or some possible extensions.

---

> ### Author Rebuttal · Authors · 2026-03-30
>
> We thank the reviewer for the thoughtful feedback and constructive suggestions.
>
> **Weakness (a).** We agree this point should be clarified. In large-scale ML, one often does not observe a comparison oracle directly, but instead has access to noisy function values from a single sample or a mini-batch. For single samples, the sign of the loss difference can be highly unreliable, so Assumption 1.1 may fail. Mini-batching can improve reliability, but typically yields gap-dependent comparison accuracy rather than a uniform margin bounded away from $1/2$. Our framework is instead motivated by settings where only comparison feedback is directly available, such as preference-based settings including RLHF. In such settings, value-based or gradient-based methods are not directly applicable, making comparison-based methods the natural object of study. We will make this distinction explicit in the revision.
>
> **Weakness (b).**
> **Motivation of the ridge model.**  The model is motivated by prior work on low-dimensional structure in high-dimensional models. Active subspace methods, for example, are based on the idea that many multivariate engineering functions vary primarily along a few directions in input space [1]. In LM fine-tuning, prior work similarly suggests that low-dimensional reparameterizations can be nearly as effective as full-space optimization [2]. We therefore adopt the ridge model as a tractable analytical abstraction of intrinsic dimension, rather than a literal claim that practical applications satisfy an exact ridge structure.
>
> [1] Constantine et al. (2014) Active subspace methods in theory and practice: applications to kriging surfaces. SIAM Journal on Scientific Computing.
>
> [2] Aghajanyan et al. (2021). Intrinsic dimensionality explains the effectiveness of language model fine-tuning. In Proceedings of the 59th annual meeting of the association for computational linguistics and the 11th international joint conference on natural language processing.
>
> **The ridge model is a strong assumption.** The current submission already includes Section 2.2 as a first step beyond exact invariance outside the active subspace. In the revision, we will strengthen this part by replacing the current subsection with a more robust extension showing that our intrinsic-dimension-related bounds can be extended to controlled non-ridge perturbations. Specifically, we consider a decomposition $f(x)=h(x)+\eta(x)$, where $h(x)=g(Ax)$, $A$ has full row rank $k$ (without loss of generality, $AA^\top=I_k$), and $\eta$ captures deviations from the exact ridge model. Thus, unlike the exact ridge setting, the objective may vary both inside and outside the $k$-dimensional subspace $\operatorname{range}(A^\top)$, provided that the unstructured component remains sufficiently dominated by the structured one.
>
> Let $B=(a_1,\ldots,a_d)$ be an orthonormal basis of $\mathbb{R}^d$ whose first $k$ vectors span $\operatorname{range}(A^\top)$, let $||\cdot||_1$ denote the $\ell_1$-norm in this basis, and let $P:=A^\top A$ be the orthogonal projector onto $\operatorname{range}(A^\top)$. We assume that $g$ is $L$-smooth, $\eta$ is differentiable, and that there exists $\beta\ge 0$ such that $||\nabla \eta(x)||_1 \le \beta ||\nabla h(x)||_1$ for all $x$, with perturbation level small enough that $\beta  ≲ 1/\sqrt{k log(d)}$. The key point is that $h$ yields the same $k$-driven descent as in the exact ridge case, while $\eta$ contributes only a dominated first-order correction and a higher-order smoothness remainder; for sufficiently small $\beta$, these do not change the intrinsic-dimension rate.  We view this as a natural strengthening of the current Section 2.2: it does not remove the fixed-subspace assumption entirely, but it does show that exact ridge structure is not required, and that controlled deviations from a reference active subspace can still preserve the intrinsic-dimension rate.
>
> **Weakness (c).** We thank the reviewer for this helpful suggestion. In the revision, we will add a short conclusion to better emphasize that the paper’s main contribution is theoretical: it develops and analyzes noisy-comparison models for smooth nonconvex zeroth-order optimization and clarifies how different noise mechanisms change the achievable rates. We will also add a limitation paragraph noting that the intrinsic-dimension bounds rely on a fixed reference low-dimensional structure, on $L$-smoothness, and on theory-driven choices of step sizes and algorithmic parameters.
>
> **Questions.** These questions are closely related to Weakness (b), where we clarify the mathematical framework and the intended scope of the linear low-dimensional structure used in the paper.

---

> > ### Author Rebuttal · Reviewer_14aE · 2026-04-03
> >
> > My main concern is Weakness (a). The authors provided some explanation on why they did not consider the stochastic setting, but they also admitted that the critical Assumption 1.1 may fail when evaluating random function values, which reveals a limitation of the proposed optimization framework.

---

> > > ### Author Response · Authors · 2026-04-03
> > >
> > > We thank the reviewer for highlighting this important modeling issue.
> > >
> > > We emphasize that Assumption 1.1 is a modeling assumption for noisy comparison feedback, intended to isolate and understand the effect of comparison noise in a clean way. This assumption has been used in prior work, for example by Saha et al. [1] in the convex and strongly convex settings, where it already leads to meaningful and nontrivial complexity guarantees. In that sense, Assumption 1.1 should not be viewed as claiming universal realism across all applications, but rather as a standard first-step abstraction that helps clarify the algorithmic difficulty induced by noisy comparisons.
> > >
> > > At the same time, we agree that in applications where comparisons are synthesized from noisy function-value evaluations, a uniform margin bounded away from $1/2$ may fail, especially near ties. This is precisely why we also introduce the confidence-oracle model in Assumption 1.2. That model is more realistic for such settings: if $f(x)=\mathbb{E}[F(x,\xi)]$ and one estimates the gap $f(x)-f(y)$ by mini-batching noisy samples, then the reliability of the induced comparison typically improves with the magnitude of the gap $|f(x)-f(y)|$, rather than being uniformly bounded away from $1/2$. Our confidence-oracle assumption is designed to capture exactly this gap-dependent behavior.
> > >
> > > More broadly, we view Assumption 1.1 and Assumption 1.2 as playing complementary roles. Assumption 1.1 provides a clean benchmark model that makes the effect of noisy comparisons analytically transparent, while Assumption 1.2 moves toward a more realistic regime in which comparison quality deteriorates near stationarity. This is analogous in spirit to classical stochastic optimization, where bounded-variance noise assumptions remain a primary analytical model for SGD even though many practical applications exhibit heavier-tailed noise. The simplified model is still valuable because it provides intuition, baseline rates, and a foundation for studying more refined noise mechanisms.
> > >
> > > [1] A. Saha, T. Koren, and Y. Mansour. Dueling Convex Optimization. ICML 2021.

---

### Decision · Program_Chairs · 2026-04-30

**Decision:**

Accept (regular)

**Comment:**

This paper studies smooth nonconvex zeroth-order optimization under noisy pairwise-comparison feedback. Its main contribution is a complexity analysis showing that, under a low-dimensional structural model, the oracle complexity depends on the intrinsic dimension k rather than the ambient dimension d. The paper also considers both a uniform-margin oracle and a more challenging gap-dependent confidence oracle. Reviewers generally agreed that the intrinsic-dimension perspective is interesting and that the theoretical direction is relevant to comparison-based optimization and preference-based learning settings.

At the same time, the reviews revealed meaningful disagreement about the overall strength of the paper. The main concerns were threefold. First, several reviewers noted that the structural assumptions are fairly strong, particularly the fixed low-dimensional/ridge-type assumption underlying the intrinsic-dimension analysis. Second, the novelty is primarily theoretical rather than algorithmic, since the search procedure itself is intentionally simple and close in spirit to classical random/direct search. Third, the initial submission had notable presentation issues, including unclear notation, delayed definitions, and insufficiently self-contained exposition.

The rebuttal addressed several of these concerns in a constructive and substantive way. In particular, the authors added a controlled synthetic study to better isolate the dependence on k, included additional empirical comparison with MeZO, and provided a more detailed discussion of robustness beyond the exact ridge setting through controlled non-ridge perturbations. The rebuttal also offered a clearer explanation of the confidence-oracle rate and the authors’ position on its tightness. These responses strengthened the paper and resolved a substantial portion of the concerns for some reviewers, although not all reservations disappeared. In particular, some concern remains about the strength and scope of the assumptions, especially the fixed low-dimensional structure used in the analysis, as well as about the overall clarity of the presentation.

On balance, I find the paper slightly above the acceptance bar. The main reason is that it makes a clear theoretical contribution to noisy comparison-based nonconvex optimization, and the rebuttal materially improved confidence in both the empirical support and the framing of the theory. That said, acceptance should be understood as recognition of a promising and technically meaningful contribution rather than as an indication that all concerns have been fully resolved.

For the final version, the paper would benefit from a careful revision of the writing and exposition. In particular, the presentation should be made more self-contained, with notation and abbreviations introduced clearly and early. Most importantly, the discussion around Assumption 1 and the low-dimensional structural model should be substantially clarified. The paper should explain more explicitly what this assumption means, why it is adopted, how it relates to intrinsic dimension in practice, and what its limitations are. It would also help to distinguish more clearly between the exact ridge setting analyzed in the paper and the broader class of practical problems that may only approximately satisfy such a structure. A more transparent discussion of this point would significantly improve the readability and interpretability of the work.